# Exact generalized Bethe eigenstates of the non-integrable alternating Heisenberg chain

Ronald Melendrez,[1, 2] Bhaskar Mukherjee,[3, 4] Marcin Szyniszewski,[4, 5]
Christopher J. Turner,[4] Arijeet Pal,[4] and Hitesh J. Changlani[1, 2]

[1] *National High Magnetic Field Laboratory, Tallahassee, Florida 32310, USA*
[2] *Department of Physics, Florida State University, Tallahassee, Florida 32306, USA*
[3] *S. N. Bose National Centre for Basic Sciences,*
*Block JD, Sector III, Salt Lake, Kolkata 700106, India*
[4] *Department of Physics and Astronomy, University College London, Gower Street, London, WC1E 6BT, UK*
[5] *Department of Computer Science, University of Oxford, Parks Road, Oxford OX1 3QD, UK*
(Dated: March 12, 2025)

Exact solutions of quantum lattice models serve as useful guides for interpreting physical phenomena in condensed matter systems. Prominent examples of integrability appear in one dimension, including the Heisenberg chain, where the Bethe ansatz method has been widely successful. Recent work has noted that certain non-integrable models harbor quantum many-body scar states, which form a superspin of regular states hidden in an otherwise chaotic spectrum. Here we consider one of the simplest examples of a non-integrable model, the alternating ferromagnetic-antiferromagnetic (bond-staggered) Heisenberg chain, a close cousin of the spin-1 Haldane chain and a spin analog of the Su-Schrieffer-Heeger model, and show the presence of exponentially many zero-energy states. We highlight features of the alternating chain that allow treatment with the Bethe ansatz (with important modifications) and surprisingly for a non-integrable system, we find simple compact expressions for zero-energy eigenfunctions for a few magnons including solutions with fractionalized particle momentum. We discuss a general numerical recipe to diagnose the existence of such generalized Bethe ansatz (GBA) states and also provide exact analytic expressions for the entanglement of such states. We conclude by conjecturing a picture of magnon pairing which may generalize to multiple magnons. Our work opens the avenue to describe certain eigenstates of partially integrable systems using the GBA.

## I. INTRODUCTION

Model spin Hamiltonians offer a minimal, yet diverse and rich platform to investigate and classify non-equilibrium dynamics of quantum many-body systems. While integrability and non-integrability of such models are currently understood to be closely tied to their athermal versus thermal behavior respectively [1–9], there are now a growing number of known systems, in arbitrary dimensions, that do not fit neatly into either category. For example, one-dimensional Rydberg atom arrays [10] and XXZ spin systems [11] have been recently shown to exhibit weak violations of ergodicity due to the presence of special states in their spectrum, which are more generally referred to as quantum many-body scars (QMBS) [12–27]. These states suggest the possible presence of "partially-integrable" manifolds [28–32] and exact or approximate conservation laws that emerge when the Hilbert space is fragmented in any spatial dimension [33–36], i.e., the Hamiltonian breaks up into variable-size Hilbert space blocks. Such atypical states and fragmented spaces are expected to be important for models with constraints, such as the PXP model [13] or icelike models of frustrated magnetism [18, 35], but they can also potentially emerge in other situations.

Two prominent features have been noted in scar-bearing models: first, they often contain a hidden or explicit "superspin" structure which is at the heart of athermal dynamics of "simple-to-prepare" initial states [10] and second, peculiarly, many models harbor a large (ex-

ponential) degeneracy at zero energy [14, 18]. While the former aspect has received considerable attention, less is known about the latter feature. At first glance, one may attribute exponentially many exact degeneracies to geometric frustration or local constraint satisfaction [37–40], however, the fact that some one-dimensional models also show such behavior [41–45] suggests this is far from the complete picture. Counts for such states have been studied with the index theorem [14, 46], and efforts to

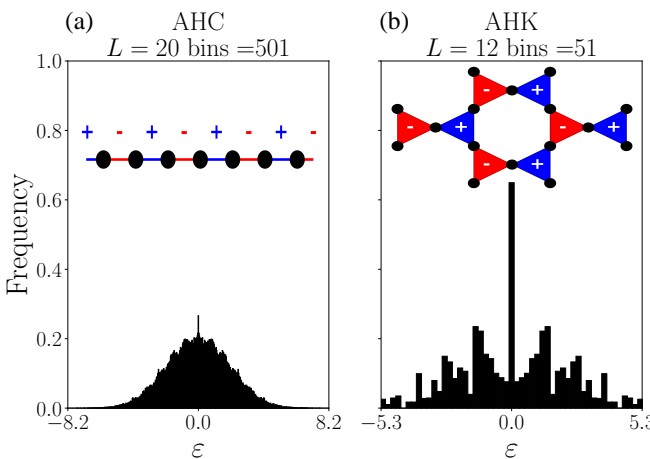

FIG. 1. Normalized density of states for alternating models with quantum many-body scars: the spin-1/2 alternating Heisenberg models on (a) a periodic chain (AHC) for $L = 20$, and (b) on the kagome lattice (AHK) for $L = 12$.

explicitly construct these states and other finite energy scar states have utilized matrix product states, [47–49], spectrum generating algebras [50], and other group theory constructions [51]. Unfortunately, a simple physical picture and the precise analytic form of these eigensolutions has remained elusive.

A notable exception is wave function ansatze (apart from matrix product states), whose form is compact and analytically tractable. One such set of wave functions is the Bethe ansatz, rooted in momentum space, which has been widely successful for translationally invariant one-dimensional integrable quantum systems. In addition to the spin-1/2 Heisenberg chain [52, 53], Bethe ansatz solvable systems include the one dimensional Hubbard model [54], Anderson impurity model [55, 56], the Kondo problem [57, 58] and models of valence fluctuations [59]. The key reason for the success of the Bethe ansatz is the tractability of the momenta of the particles in an integrable model – when two particles collide, their momenta after the collision are simply exchanged. In contrast, for a non-integrable model, a scattering process typically allows more general momenta subject only to momentum conservation. Since this admits a large phase space of possibilities, the wave functions are expected to be (generally) unstructured – which is the essence of quantum chaotic behavior. Thus it is nontrivial that there exist non-integrable models where the Bethe ansatz should apply at all [25, 60, 61]. When viewed from the lens of Hilbert space fragmentation however, it is conceivable that a non-integrable model may contain states which mix only with a few other states in the Hilbert space, and thus some compact efficient representation of certain eigenstates may be possible [62].

The purpose of the present work is to show that a generalized Bethe ansatz (GBA) yields exact, analytic (closed form) expressions of some eigenstates of arbitrary finite length chains, in one of simplest non-integrable models in one dimension – the spin-1/2 alternating Heisenberg chain (AHC). The alternation/staggering can also be introduced for high-dimensional systems, for example, on the alternating Heisenberg kagome (AHK) the up triangles have ferromagnetic interactions while the down triangles have antiferromagnetic interactions. Both models were introduced previously [18] as platforms for QMBS with an embedded superspin at zero energy. The many-body density of states (plotted across all $S_z$ sectors combined) for both AHC and AHK has a peak at zero energy, as shown in Fig. 1. This feature is typically absent in non-integrable models and suggests the importance of the model's symmetry properties.

Motivated by this spectral structure, we have carried out further investigations of the AHC model. Our research builds on the work of Ref. [63] that laid out a Bethe ansatz recipe to determine two-magnon states, at all energies for a more general version of the AHC (which we clarify shortly). The solutions can be written as a self-consistent set of equations that must be solved to determine the parameters entering the ansatz, including

the generalized momenta. We applied this recipe, with modifications, which we refer to as the GBA, to the AHC model on a finite arbitrary-length chain, targeting zero energy states. As a result, we obtained exact closed-form expressions for arbitrary system sizes and analytically explained the finite-size even-odd dependence of the zero energy degeneracies, as depicted in Table I. Importantly, we have also extended the GBA approach to three magnons, providing exact expressions and explanations for the zero energy state counts.

These analytic solutions provide a physical picture of how magnons organize themselves in the AHC model at zero energy, by either effectively avoiding each other, or destructively interfering to avoid the Ising penalty. Inspired by this, we have also provided a numerical recipe for the GBA which allows us to show that partial integrability of the AHC is present for a higher number of magnons. The observed patterns lead us to conjecture that a large number of magnons may "pair up" to form zero energy states as well.

The rest of the paper is organized as follows. In Sec II we formally introduce the AHC and briefly discuss its one-magnon dispersion. Then in Sec. III we discuss the two-magnon Schrödinger equation and solve it both by physical reasoning and more formally via the GBA. In Sec. IV we use this experience to obtain eigenstates of the three-magnon problem. In Sec. V we show the results of the numerical Bethe ansatz, which allows us to address the case of four magnons to explore how magnons pair up. In Sec. VI we discuss the entanglement properties of the Bethe ansatz states which we find to be area-law entangled. We conclude in Sec. VII by summarizing our findings and discussing the implications of our results – in particular, we find that our results provide a picture of how "few-body" quantum scars emerge in the AHC (and related) models.

## II. MODEL HAMILTONIAN AND SINGLE-MAGNON DISPERSION

The Hamiltonian of the spin-1/2 alternating Heisenberg chain (AHC) is,

$$H = \sum_{i=0}^{2N-1} (-1)^i \, \mathbf{S}_i \cdot \mathbf{S}_{i+1}, \tag{1}$$

with $L = 2N$ sites and where $i$ is a site index, $i + 1$ is the nearest neighbor taken modulo $2N$ for periodic boundary conditions. $S_i^\alpha = \frac{1}{2}\sigma_i^\alpha$, where $\alpha \in x, y, z$ denotes spin-1/2 operators. Due to the alternating sign, the superspin is embedded as a ferromagnetic multiplet at exactly zero energy. This model is a magnetic analog of the fermionic Su-Schrieffer Heeger chain [64] and it is a special case of

| | N | | | | | | | Unit Cells | | | | | | | $C_{N_\downarrow}$ | |
|---|---|---|---|---|---|---|---|---|---|---|---|---|---|---|---|---|
| $N_\downarrow$ | 4 | 5 | 6 | 7 | 8 | 9 | 10 | 11 | 12 | 13 | 14 | 15 | 16 | even $N$ | odd $N$ |
| 1 | 2 | 2 | 2 | 2 | 2 | 2 | 2 | 2 | 2 | 2 | 2 | 2 | 2 | $C_1 = 2$ | $C_1 = 2$ |
| 2 | 4 | 9 | 6 | 13 | 8 | 17 | 10 | 21 | 12 | 25 | 14 | 29 | 16 | $C_2 = N$ | $C_2 = 2N - 1$ |
| 3 | 10 | 16 | 10 | 24 | 14 | 32 | 18 | 40 | 22 | 48 | 26 | 56 | 30 | $C_3 = 2(N-1)$ for $N > 4$ | $C_3 = 4(N-1)$ |
| 4 | 10 | 26 | 17 | 51 | 28 | 84 | 49 | 125 | 66 | 174 | 97 | 231 | 120 | $C_4 = \frac{N^2-2}{2}$ for $4 \nmid N$; $C_4 = \frac{N(N-1)}{2}$ for $4 \mid N$ | $C_4 = N^2 + \frac{N-3}{2}$ |
| 5 | – | 36 | 24 | 78 | 42 | 136 | 80 | 210 | 110 | 300 | 168 | 406 | 210 | $C_5 = 2C_4 - C_3$ | $C_5 = 2C_4 - C_3$ |
| 6 | – | – | 26 | 113 | 56 | 238 | 128 | 435 | 220 | 718 | 376 | 1103 | 560 | | $C_6 = \binom{N}{3}$ for $4 \mid N$ |

TABLE I. Exact diagonalization results for the number of zero energy states ($C_{N_\downarrow}$) as a function of the number of unit cells $N$ and magnon number $N_\downarrow$. The formulae for $C_{N_\downarrow}$ are obtained from numerical inference. $4 \mid N$ means $N$ is divisible by 4.

a more general family of AHC-XXZ models,

$$H = \sum_{n=0}^{N-1} \frac{J_1}{2} \left(S_{2n}^+ S_{2n+1}^- + S_{2n}^- S_{2n+1}^+\right) + \Delta_1 S_{2n}^z S_{2n+1}^z$$
$$+ \sum_{n=0}^{N-1} \frac{J_2}{2} \left(S_{2n}^+ S_{2n-1}^- + S_{2n}^- S_{2n-1}^+\right) + \Delta_2 S_{2n}^z S_{2n-1}^z, \quad (2)$$

where $\Delta_1$ and $\Delta_2$ have been introduced to allow for anisotropy in the spin-spin coupling, for each of the kinds of bonds. When these Ising terms are absent ($\Delta_1 = \Delta_2 = 0$) the model maps to spinless fermions via Jordan Wigner transformation and becomes integrable [65]. The ground states of the spin-1/2 AHC ($J_1 = \Delta_1 = -J_2 = -\Delta_2 = 1$) and the spin-1 antiferromagnetic Heisenberg model ($J_1 = \Delta_1 \to -\infty$ and $J_2 = \Delta_2 = 1$) are adiabatically connected [66–71], consequently on open chains, the AHC has a nearly degenerate 4 fold gapped spectrum for even $N$. We also note that the model has a long history of practical relevance, for example, the study of alternating chains has enabled an explanation of magnetic ordering of various chemical compounds [72–75].

The AHC is non-integrable and belongs to the Gaussian Orthogonal Ensemble random matrix class [76]. However, this does not exclude the possibility of integrable sectors in the model, as mentioned earlier, it is now understood that few-magnon sectors may be amenable to an exact Bethe ansatz treatment [63]. We will see explicitly later why this is the case.

Consider first the case of a single magnon in a ferromagnetic background $|F\rangle \equiv |\uparrow\uparrow \dots \uparrow\rangle$. The exact wave function for this sector reads

$$|\psi_{1mag}\rangle = \sum_{n=0}^{N-1} a_{2n}|2n\rangle + a_{2n+1}|2n+1\rangle, \quad (3)$$

where $2n$ and $2n+1$ denote even ($A$) and odd ($B$) enumerated sites on the lattice that contains a flipped spin in an otherwise ferromagnetic background i.e. $|i\rangle \equiv S_i^- |F\rangle$. Our task is to obtain the functional form of $a$'s that satisfy the Schrödinger equation $H|\psi_{1mag}\rangle = \varepsilon|\psi_{1mag}\rangle$. (We will generally not be too concerned about normalizing the states especially when we consider higher-magnon

states.) Plugging the AHC-XXZ Hamiltonian into this equation gives,

$$J_1 a_{2n+1} + J_2 a_{2n-1} = 2E a_{2n}, \quad (4a)$$
$$J_1 a_{2n} + J_2 a_{2n+2} = 2E a_{2n+1}, \quad (4b)$$

where we have defined $E \equiv \varepsilon - \frac{(\Delta_1+\Delta_2)}{4}(N-2)$. Plane waves solve the above equations, for which we parameterize the coefficients as $a_{2n} = \alpha_A e^{2ikn}$ and $a_{2n+1} = \alpha_B e^{ik(2n+1)}$ which gives the two equations,

$$\left(J_1 e^{+ik} + J_2 e^{-ik}\right)\alpha_B = 2E\alpha_A, \quad (5a)$$
$$\left(J_1 e^{-ik} + J_2 e^{+ik}\right)\alpha_A = 2E\alpha_B. \quad (5b)$$

The solution gives two bands with energies,

$$E_\pm = \pm \frac{\sqrt{J_1^2 + J_2^2 + 2J_1 J_2 \cos 2k}}{2}, \quad (6)$$

and eigenvector coefficients,

$$\left(\frac{\alpha_B}{\alpha_A}\right)_\pm = \pm\sqrt{\frac{J_1 e^{-ik} + J_2 e^{ik}}{J_1 e^{ik} + J_2 e^{-ik}}}. \quad (7)$$

For the AHC ($J_1 = \Delta_1 = -J_2 = -\Delta_2 = 1$) we get,

$$\varepsilon_\pm = E_\pm = \pm \sin k \quad \text{and} \quad \left(\frac{\alpha_B}{\alpha_A}\right)_\pm = \pm i. \quad (8)$$

Using $\alpha_A = 1$ and incorporating the appropriate normalization, one can see that the eigenmodes are analogous to "right circularly" or "left circularly" polarized states,

$$S_{E=\pm \sin(k)}^- = \frac{1}{\sqrt{2}} S_{k,A}^- \pm \frac{i}{\sqrt{2}} S_{k,B}^-, \quad (9)$$

where we have defined the Fourier transforms on the individual even and odd sublattices as,

$$S_{k,A(B)}^- = \frac{1}{\sqrt{N}} \sum_{n=0}^{N-1} e^{ik(2n(+1))} S_{2n(+1)}^-. \quad (10)$$

Clearly, there are two zero energy states ($\varepsilon = E = 0$) for the case of $k = 0$, one for each band. Due to this degeneracy, $S_{k=0,A}^- |F\rangle$ and $S_{k=0,B}^- |F\rangle$ are also zero energy states. These correspond to the magnon being in a

uniform superposition purely on sublattice $A$ or purely on sublattice $B$. Alternatively one could view the two one-magnon states as $S_{\text{tot}}^- |F\rangle = (S_{k=0,A}^- + S_{k=0,B}^-) |F\rangle$, which we refer to as the "uniform mode", and $(S_{k=0,A}^- - S_{k=0,B}^-) |F\rangle$, the "staggered mode". While the uniform zero mode is a simple consequence of SU(2) symmetry, the existence of the staggered zero mode is specific to the AHC.

## III.   TWO-MAGNON ZERO ENERGY STATES

The one-magnon band structure discussed in the previous section raises an intriguing physical possibility. Can one construct the two-magnon zero energy states by combining one-magnon states with opposite energies? If magnons were truly non-interacting, one could simply add up their individual energies and make a two-magnon wave function which was a simple product state. The zero energy condition could be achieved by taking two magnons in the same band with opposite momenta $k, -k$, or one magnon in each band with the same momentum $k$, or alternatively, both magnons in individual one-magnon zero energy states. It turns out that this way of combining individual magnons, while correct in spirit, does not account for the fact that magnons collide and interact.

In this section, we address this problem and arrive at exact two-magnon eigensolutions. Working with the AHC Hamiltonian ($J_1 = \Delta_1 = -J_2 = -\Delta_2 = 1$), we provide the explicit parameterization of each family of two-magnon zero energy states for arbitrary $N$, and provide a rigorous proof for their linear independence, which is essential for explaining counts seen in exact diagonalization (Table I). We also note that the GBA parametrization presented in this work is complementary to the real space picture previously discussed by us elsewhere [51]. In that picture, an exact basis was constructed using states where two magnons are placed a fixed distance apart from one another.

We adopt the notation, with loss of generality for $i < j$, $|i\ j\rangle \equiv S_i^- S_j^- |F\rangle$. Since the two magnons can either be on the same or different sublattices, there are four types of coefficients in the two-magnon wave function,

$$|\psi_{2mag}\rangle = \sum_{i<j} a_{i,j} |i,j\rangle. \tag{11}$$

Analogous to the case of one magnon, we apply the Hamiltonian [Eq. (2)] to the two-magnon state in Eq. (11) and solve the corresponding eigenvalue equation $H|\psi_{2mag}\rangle = \varepsilon|\psi_{2mag}\rangle$. We define $E \equiv \varepsilon - (\Delta_1 + \Delta_2)\frac{(N-4)}{4}$. When magnons are not nearest neighbors we

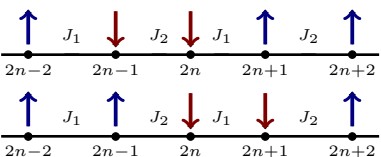

FIG. 2. Configurations for the constraint equations for two magnons.

get,

$$2Ea_{2n,2m} = J_1(a_{2n,2m+1} + a_{2n+1,2m}) + $$
$$J_2(a_{2n-1,2m} + a_{2n,2m-1}), \tag{12a}$$
$$2Ea_{2n+1,2m+1} = J_1(a_{2n,2m+1} + a_{2n+1,2m}) + $$
$$J_2(a_{2n+2,2m+1} + a_{2n+1,2m+2}), \tag{12b}$$
$$2Ea_{2n,2m+1} = J_1(a_{2n,2m} + a_{2n+1,2m+1}) $$
$$+ J_2(a_{2n-1,2m+1} + a_{2n,2m+2}), \tag{12c}$$
$$2Ea_{2n-1,2m} = J_1(a_{2n-2,2m} + a_{2n-1,2m+1}) $$
$$+ J_2(a_{2n,2m} + a_{2n-1,2m-1}). \tag{12d}$$

We collectively refer to these as the "hopping" equations.

When magnons are nearest neighbors, spin exchange processes are disallowed and we get,

$$2(E - \Delta_2)a_{2n-1,2n} = J_1 a_{2n-2,2n} + J_1 a_{2n-1,2n+1}, \tag{13a}$$
$$2(E - \Delta_1)a_{2n,2n+1} = J_2 a_{2n-1,2n+1} + J_2 a_{2n,2n+2}, \tag{13b}$$

which we refer to as "constraint" equations.

From here on we focus on the AHC, i.e. $J_1 = \Delta_1 = -J_2 = -\Delta_2 = 1$. For this choice, the middle of the spectrum corresponds to $\varepsilon = E = 0$. When written compactly, the four hopping equations now read,

$$0 = + (-1)^n a_{n+1,m} + (-1)^m a_{n,m+1} $$
$$- (-1)^n a_{n-1,m} - (-1)^m a_{n,m-1}, \tag{14}$$

and the constraint equations are,

$$2a_{n,n+1} = a_{n-1,n+1} + a_{n,n+2}. \tag{15}$$

A simple solution to the hopping and constraint equations can be seen right away – it is $a_{ij} = $ constant for any $i, j$. This should not be surprising, the solution corresponds to the case $(S_{\text{tot}}^-)^2 |F\rangle$, the two-magnon uniform mode, which is part of the degenerate multiplet associated with the ferromagnet. But there are many more solutions to the six equations, we discuss next a strategy for deriving these and then formalize our notions with the GBA.

### A.   Solutions to the hopping equations

We first determine the solutions of the hopping equations alone. (These provide a basis of functions that will

be linearly combined in a bid to satisfy the additional constraint equations.) These are given by a product of two plane waves,

$$a_{n,m} = \alpha_{f(n),f(m)} e^{i(k_1 n + k_2 m)}, \qquad (16)$$

with $n < m$ and where $f(i) = A$ for $i$ even and $f(j) = B$ for $j$ odd, this is a function that indicates the sublattice of a site. Plugging in this form into the hopping equations [Eqn. (14)] gives four equations, which we write in matrix-vector form as,

$$\begin{pmatrix} \sin k_2 & -\sin k_1 & 0 & 0 \\ -\sin k_1 & \sin k_2 & 0 & 0 \\ 0 & 0 & \sin k_2 & \sin k_1 \\ 0 & 0 & \sin k_1 & \sin k_2 \end{pmatrix} \begin{pmatrix} \alpha_{AA} \\ \alpha_{BB} \\ \alpha_{AB} \\ \alpha_{BA} \end{pmatrix} = \begin{pmatrix} 0 \\ 0 \\ 0 \\ 0 \end{pmatrix}. \qquad (17)$$

For this set of four equations to have a non-trivial solution, we need the determinant of the matrix in Eq. (17) to be zero. This results in the condition $\sin^2 k_1 = \sin^2 k_2$ which is equivalent, not surprisingly, to the sum of non-interacting energies $\sin k_1 \pm \sin k_2 = 0$.

There are multiple solutions. For $k_1, k_2$ real we get,

$$k_1 = \pm k_2 \quad \text{and} \quad k_1 = \pi \pm k_2, \qquad (18)$$

each of which we must review separately when we attempt to satisfy the constraint equations. Since the shifts of $\pi$ in the momenta can be absorbed into the $\alpha_{f(n),f(m)}$ Bethe coefficients, they do not constitute new cases i.e. linearly independent functions, and are hence not considered. We note that $k_i$ can be complex-valued, as is the case of bound states described by the Bethe ansatz. However, we work with $k_1, k_2$ real as these will be found to account for all zero energy states in the two-magnon sector.

### B. Exact zero-energy solutions for arbitrary $N$

We will now see that the exact zero-energy eigenstates, those that satisfy both hopping and constraint equations, emerge rather naturally for the case where the magnons have net zero center of mass momentum ($k_1 = -k_2$). For even $N$ we will find that *all*, and for odd $N$ *about half* the zero energy solutions satisfy this momentum condition.

When $k_1 = -k_2$ (and not zero) we obtain from the hopping equations, $\alpha_{AA} = -\alpha_{BB}$ and $\alpha_{AB} = \alpha_{BA}$. (When $k_1$ and $k_2$ are zero, there is no restriction on the $\alpha$'s at the level of the hopping equations, and this case will be reviewed separately.) Consider the possibility that $a_{2n,2m+1} = a_{2n-1,2m} = 0$ for any $n, m$ i.e. $\alpha_{AB}$ the amplitude of having the two magnons on opposite sublattices is zero. With this condition, simple inspection reveals that the constraint equations, Eqs. (15), can also be satisfied by demanding that $a_{i,j}$ depends only on the separation $|i-j|$ (i.e. the distance between the magnons). This requires that the amplitude of both magnons being on $A$ is the negative of the amplitude of both magnons on

$B$. Effectively, the magnons avoid the constraint of being nearest neighbors by always being on the same sublattice.

Periodic boundary conditions impose the relation on the coefficients,

$$a_{i,j} = a_{j,i+2N}, \qquad (19)$$

for $i, j$ both even or both odd. Since we expect periodic functions, we attempt a solution with cosine functions and get,

$$a_{2n,2m} = \cos(k(2n - 2m)) = \cos(k(2m - 2n) - 2kN), \qquad (20)$$

which implies that $k = p\pi/N$, where $p$ is an integer. For this choice of $k$ we have,

$$a_{2n,2m} = -a_{2n+1,2m+1} = +\cos(k(2n - 2m)), \quad (21a)$$
$$a_{2n,2m+1} = a_{2n-1,2m} = 0. \qquad (21b)$$

We refer to this set as "$AA - BB$ solutions". For even $N$, we take our linearly independent set to be indexed by $p = 0, 1, 2, ..., (N/2 - 1)$. It is straightforward to see that $p' = N - p$ gives the same physical wave function and thus $p > N/2$ is not considered. However, note that we also eliminated $p = N/2$, i.e. $k = \pi/2$ in our linearly independent set. We clarify this with the help of a derivation in Appendix A, where we show that the $p = N/2$ solution can be written as a linear combination of the other $N/2$ states. Thus for even $N$ there are $N/2$ linearly independent $AA - BB$ solutions. For odd $N$, we take $p = 0, 1, 2, ..., (N-1)/2 - 1$, since there are $(N-1)/2$ $AA - BB$ solutions by the same argument.

One could have alternatively worked with purely sine functions and obtained

$$a_{2n,2m} = \sin(k(2n-2m)) = \sin(k(2m-2n) - 2kN), \quad (22)$$

which would imply that $k = (2p+1)\pi/2N$, i.e., a different quantization condition for $k$, which gives,

$$a_{2n,2m} = -a_{2n+1,2m+1} = \sin(k(2n - 2m)), \quad (23a)$$
$$a_{2n,2m+1} = a_{2n-1,2m} = 0. \qquad (23b)$$

This solution set is not linearly independent of the cosine series. (We discuss the numerical prescription for checking the linear dependence in Sec. III F.)

Another class of solutions can be obtained by demanding that $a_{2n,2m} = a_{2n+1,2m+1} = 0$ for all $n, m$ i.e. the amplitude is non-zero only when the two magnons are on opposite sublattices. In this case, Eqs. (14) are trivially satisfied. The two constraint equations become,

$$2a_{2n-1,2n} = 0, \qquad 2a_{2n,2n+1} = 0. \qquad (24)$$

Thus, we require the wave function to vanish when the two magnons are nearest neighbors. Unlike the previous case, a simple cosine or sine will not do the trick unless $k$ has a special value. We work around this by using certain freedom in the hopping equations, note that they can be satisfied even if $a_{ij}$ is shifted by a constant shift

i.e. $a_{ij} \to a_{ij} + C$. This constant $C$ is adjusted so that the wave function vanishes for nearest neighbor magnons. We arrive at the set of zero energy solutions, which we refer to as "$AB$ solutions",

$$a_{2n,2m} = a_{2n+1,2m+1} = 0, \qquad (25a)$$

$$\begin{aligned} a_{2n,2m+1} &= a_{2n-1,2m} \\ &= \cos(k(2n-2m-1)) - \cos k, \qquad (25b) \end{aligned}$$

where $k = p\pi/N$ and $p$ is an integer. $k = 0$ and $k = \pi/2$ are not allowed since the wave function would vanish everywhere for those choices. Thus, for even $N$, $p$ is $1, 2, 3, ..., (N/2 - 1)$, i.e. there are $(N/2 - 1)$ solutions. For odd $N$, $k = \pi/2$ is not an allowed momentum and $p = 1, 2, 3, ..., (N-1)/2$, i.e. there are $(N-1)/2$ solutions.

For even $N$, the two families of solutions and the uniform mode, give a total number of $N/2 + N/2 - 1 + 1 = N$ linearly independent solutions, perfectly consistent with Table I. For odd $N$, we may (naively) conclude that the total number of solutions is only $(N-1)/2 + (N-1)/2 + 1 = N$. However, there are $N - 1$ zero energy states still missing when compared to those observed in Table I. We address this puzzle in Sec. III D.

The momentum zero solutions can be understood from the point of view of the symmetries of the Hamiltonian, which we have not explicitly utilized up to this point. Denoting translation by a single lattice constant to be $\tau$, the Hamiltonian with periodic boundary conditions is invariant under translation by two lattice constants, i.e. $\tau^2$. In the net momentum zero sector we have,

$$\tau^2|\psi_{2mag}\rangle = |\psi_{2mag}\rangle \implies \tau|\psi_{2mag}\rangle = \pm|\psi_{2mag}\rangle. \quad (26)$$

For the case of net momentum zero, translating by one lattice constant is identical to reflection about an axis that passes through an arbitrary bond on the lattice, we thus refer to this eigenvalue as $R = +1$ or $R = -1$. The $AA - BB$ solutions satisfy $\tau|\psi_{2mag}\rangle = -|\psi_{2mag}\rangle$ ($R = -1$) and the $AB$ solutions satisfy $\tau|\psi_{2mag}\rangle = +|\psi_{2mag}\rangle$ ($R = 1$). The uniform mode also satisfies $\tau|\psi_{2mag}\rangle = |\psi_{2mag}\rangle$ ($R = 1$). For even $N$, the total number of zero energy solutions is the same in $R = +1$ and $R = -1$ sectors. This symmetry classification is useful not only from the point of view of providing some insight into the nature of the solutions, but also helps reduce the number of parameters we need to deal with.

### C. Generalized Bethe ansatz approach

We now arrive at the same results of the previous section in a more formal way our employing the GBA – we use this nomenclature to refer to any linear combination of the usual Bethe ansatz states appropriately incorporating sublattice labels. For example, we consider,

$$\begin{aligned} a_{n,m} &= \alpha_{f(n),f(m)} e^{i(k_1 n + k_2 m)} \\ &+ \beta_{f(n),f(m)} e^{i(k_2 n + k_1 m)} + C_{f(n),f(m)}, \qquad (27) \end{aligned}$$

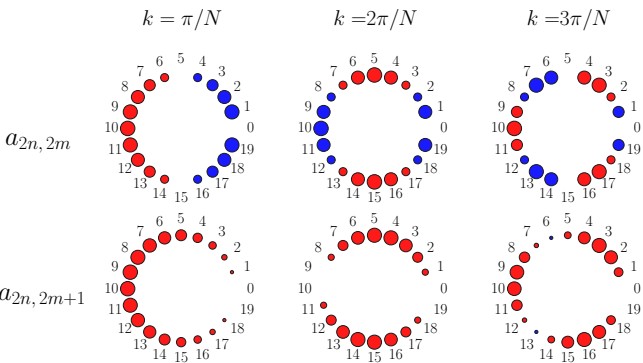

FIG. 3. Conditional probability amplitude for Bethe coefficients. Top row: $AA - BB$ solutions $a_{2n,2m} = \cos(k(2n - 2m))$. Bottom row: $AB$ solutions $a_{2n,2m+1} = \cos(k(2n - 2m - 1)) - \cos(k)$. Calculations for unit cell $N = 20$ with the first magnon fixed at $n = 0$ and $m$ varied around the circle. Dot size represents amplitude strength, with blue denoting positive and red negative values. Momentum values are specified above each figure.

where we have added constant terms that are not usually present in the Bethe ansatz. This is equivalent to taking a linear combination of basis functions, one with momenta $k_1, k_2$ and the other with $0, 0$. In principle, one could use more than two basis functions and with momenta that themselves are completely general i.e. unspecified to begin with.

Consider the case of $k_1 = -k_2 \equiv k$. Similar to what we saw before, plugging in the ansatz into the hopping equations and then applying periodic boundary conditions gives, $\alpha_{AA} = -\alpha_{BB}$, $\beta_{AA} = -\beta_{BB}$, $\alpha_{AB} = \alpha_{BA}$, and $\beta_{AB} = \beta_{BA}$. There is no restriction on the constant terms from the hopping equations alone.

The number of free parameters is further reduced by the use of periodic boundary conditions on the coefficients. The even-even and odd-odd coefficients map to themselves, while the even-odd and odd-even coefficients map onto one another. Using these conditions we get,

$$\frac{\beta_{AA}}{\alpha_{AA}} = e^{-2ikN}, \quad \frac{\beta_{AB}}{\alpha_{AB}} = e^{-2ikN}, \quad C_{AB} = C_{BA}. \quad (28)$$

Using these to simplify the wavefunction Ansatz,

$$\begin{aligned} a_{2n,2m} &= \alpha_{AA} e^{ik(2n-2m)} \\ &+ \alpha_{AA} e^{-2ikN} e^{-ik(2n-2m)} + C_{AA}, \qquad (29a) \end{aligned}$$

$$\begin{aligned} a_{2n,2m+1} &= \alpha_{AB} e^{ik(2n-2m-1)} \\ &+ \alpha_{AB} e^{-2ikN} e^{-ik(2n-2m-1)} + C_{AB}, \quad (29b) \end{aligned}$$

$$\begin{aligned} a_{2n-1,2m} &= \alpha_{AB} e^{ik(2n-2m-1)} \\ &+ \alpha_{AB} e^{-2ikN} e^{-ik(2n-2m-1)} + C_{AB}, \quad (29c) \end{aligned}$$

$$\begin{aligned} a_{2n+1,2m+1} &= -\alpha_{AA} e^{ik(2n-2m)} \\ &- \alpha_{AA} e^{-2ikN} e^{-ik(2n-2m)} + C_{BB}. \qquad (29d) \end{aligned}$$

When we impose $R = 1$, it immediately follows that $\alpha_{AA} = 0$ and $C_{BB} = C_{AA}$. With these conditions, the two constraint equations give the same resultant equation

$$2(\alpha_{AB}e^{-ik} + \alpha_{AB}e^{-2ikN}e^{ik} + C_{AB}) = 2C_{AA}. \quad (30)$$

Setting $\alpha_{AB} = 1$, since it is just a scale factor, and choosing $2kN = 2p\pi$ where $p$ is an integer we get,

$$C_{AA} - C_{AB} = \cos k. \quad (31)$$

Since only the difference $C_{AA} - C_{AB}$ is constrained, we are free to choose $C_{AA} = 0$ and $C_{AB} = -\cos k$. Assembling all our results we recover the $AB$ solution set as in Eqs. (25).

For $R = -1$, we get $\alpha_{AB} = 0$ and $C_{AB} = 0$ and $C_{BB} = -C_{AA}$. With these conditions, the two constraint equations give the same equation

$$0 = (-\alpha_{AA}e^{-2ik} - \alpha_{AA}e^{2ik}e^{-2ikN} - C_{AA})$$
$$+ (\alpha_{AA}e^{-2ik} + \alpha_{AA}e^{2ik}e^{-2ikN} + C_{AA}), \quad (32)$$

which holds for arbitrary $\alpha_{AA}$, $k$ and $C_{AA}$ without any restrictions. Choosing $2kN = 2p\pi$ and $\alpha_{AA} = 1$ (scale factor) and $C_{AA} = 0$ we recover the $AA - BB$ set of solutions as in Eqs. (21). In Appendix B we also check the case of $k_1 = k_2 \neq 0$ and find that it (alone) can not yield an exact zero-energy solution.

### D. Additional exact zero energy solutions for odd $N$ with "fractionalized" momenta

For odd $N$, we find that superposition of momentum pairs $(k, k)$ and $(k + \pi/2, k + \pi/2)$ also yield exact $E = 0$ eigensolutions. For even $N$, $e^{2ikN} = e^{2i(k+\pi/2)N} = 1$, while for odd $N$ $e^{2ikN} = 1$ and $e^{2i(k+\pi/2)N} = -1$ i.e. $k + \pi/2$ is a "fractionalized" momentum for odd $N$. This difference between even and odd $N$ manifests itself as extra zero energy states for odd $N$.

We find that the extra family of zero energy states is given by,

$$a_{2n,2m} = e^{ik(2n+2m)}, \quad (33a)$$
$$a_{2n,2m+1} = e^{i(k+\pi/2)(2n+2m+1)}(-i\cos k), \quad (33b)$$
$$a_{2n-1,2m} = e^{i(k+\pi/2)(2n+2m-1)}(+i\cos k), \quad (33c)$$
$$a_{2n+1,2m+1} = e^{ik(2n+2m+2)}. \quad (33d)$$

Note that momentum $k$ is present only in the $AA$ and $BB$ coefficients and momentum $k + \pi/2$ is present only in the $AB$ and $BA$ coefficients. While we have simply stated the solution here, we will revisit how these coefficients arise in the context of three magnons (see Appendix C).

It is straightforward to check that the hopping and constraint equations are satisfied by these wave function coefficients, irrespective of whether $N$ is even or odd. However, the coefficients also need to satisfy periodic boundary conditions. For $e^{2ikN} = 1$, we find that $a_{2n,2m} =$

$a_{2m,2n+2N}$ and $a_{2n+1,2m+1} = a_{2m+1,2n+1+2N}$ hold for arbitrary $N$. But the requirement that $a_{2n,2m+1} = a_{2m+1,2n+2N}$ implies

$$(-1)^{n+m}e^{ik(2n+2m+1)}\cos k$$
$$= (-1)^{m+1+n+N}e^{ik(2n+2m+1+2N)}\cos k, \quad (34)$$

which holds only for odd $N$, or $k = \pi/2$, or both.

Consider the possibility $k = \pi/2$ first, which is an allowed momentum only for even $N$. For this case, we recover the $AA - BB$ solution, accounted for earlier. To see this, note that $e^{i(\pi/2)(2n+2m)} = e^{i(\pi/2)(2n-2m)}$. The $\cos(\pi/2)$ factor ensures that there is no weight on $AB/BA$ coefficients.

Thus, our proposed solution set yields new exact zero energy states only for odd $N$. The number of solutions is the number of allowed $k = p\pi/N$, which for odd $N$ we take to correspond to $p = 0, 1, 2, 3, ..., (N-1)$. (This series can not give $k = \pi/2$ for odd $N$. Also $p = 0$ i.e. $k = 0$ and $p = N$ i.e. $k = \pi$ correspond to the same solution.) Thus, we have additional $N$ solutions in this family.

However, the uniform mode can be written as a linear combination of this family and the $AB$ solutions and thus the number of additional linearly independent states is $N-1$. This brings the total number of zero energy states, for odd $N$, to be $N + (N-1) = 2N - 1$. This resolves the mystery of the extra solutions for odd $N$ and explains exactly what we find in Table I.

### E. Generalization to the XXZ case

Till this point, we have focused on the isotropic Heisenberg case. It is thus natural to ask how our results generalize to the XXZ case. Observe that the hopping equations, up to an Ising shift which vanishes entirely for $\Delta_1 = -\Delta_2$, are unaffected by the choice of anisotropy. When it comes to the constraint equations, the two-magnon wave function, for both $AA - BB$ and $AB$ solutions, vanishes for nearest neighbors i.e. $a_{2n-1,2n} = a_{2n,2n+1} = 0$. Thus the value of the anisotropy, as long as $\Delta_1 = -\Delta_2$ is irrelevant for both families of $E = 0$ solutions.

However, there is one solution, of the $R = 1$ type, which is sensitive to the value of $\Delta_1$ and $\Delta_2$. This is the uniform mode i.e. $a_{i,j} = 1$. As mentioned earlier, this two-magnon wave function is $(S_{\text{tot}}^-)^2 |F\rangle$ and it must be degenerate with the ferromagnetic state for the SU(2) symmetric case since it is part of the multiplet of maximal spin. But by adding in an anisotropy, the symmetry is broken and this mode is no longer an exact eigensolution.

The generalized version of this mode for the case of $\Delta_1 = -\Delta_2 = \Delta$ and $J_1 = -J_2 = 1$ is straightforward to obtain. For $E = 0$ the constraint equations read,

$$2\Delta\, a_{2n-1,2n} = a_{2n-2,2n} + a_{2n-1,2n+1}, \quad (35a)$$
$$2\Delta\, a_{2n,2n+1} = a_{2n-1,2n+1} + a_{2n,2n+2}. \quad (35b)$$

We observe, by simple inspection, that the four hopping and two constraint equations are satisfied by choosing,

$$a_{2n,2m} = a_{2n+1,2m+1} = \Delta, \qquad (36a)$$

$$a_{2n,2m+1} = a_{2n-1,2m} = 1. \qquad (36b)$$

Not surprisingly, we recover the uniform mode as a special case in the Heisenberg limit, $\Delta = 1$.

Let us take a closer look at the case of $\Delta = 0$. In this case, there are no "interactions" and the model becomes completely exactly solvable via mapping to free fermions using a Jordan Wigner transformation [65]. In terms of the language presented here, we can clearly see that some constraints are lifted. More precisely, note that the constraint equations relate the $AB$ and $BA$ coefficients to the $AA$ and $BB$ coefficients. However, for $\Delta = 0$ the $AB$ and $BA$ coefficients couple to one another only via the hopping equations and do not appear in the constraint equations. The $AA$ and $BB$ coefficients couple to one another in the hopping and two constraint equations. Thus the decoupling of the $AB/BA$ and $AA/BB$ coefficients leads to an increased number of two-magnon $E = 0$ states for the case of $\Delta = 0$.

### F. Numerical checks on the number of linearly independent solutions

The two-magnon eigenfunctions, while compact in their representation, are not guaranteed to be orthogonal to one another, and hence are not necessarily linearly independent. We check for their linear independence by normalizing the analytic solutions numerically, and then computing the overlap (Gram) matrix,

$$S_{ij} = \langle \psi_i^a | \psi_j^a \rangle = S_{ji}^*. \qquad (37)$$

We numerically diagonalize this Hermitian matrix to check its rank (and more generally, the distribution of eigenvalues). The eigenvalues (when sorted from largest to smallest) show a step, which in turn yields the number of linearly independent states. The scheme was useful as it guided the search for appropriate mathematical identities that explicitly (and rigorously) helped clarify why certain states were linearly dependent on others. The linearly independent states can also be constructed via a Gram-Schmidt orthogonalization procedure coupled with a basis reduction step.

## IV. THREE-MAGNON ZERO ENERGY STATES

We now consider the case of three-magnon zero energy states. Some states are easy to explicitly write down

– they are obtained by acting $S_{\text{tot}}^-$ on the two-magnon states we derived in the previous section. However, Table I reveals that the number of zero energy states is almost doubled with respect to the two-magnon case, for both even and odd $N$. The objective of this section is

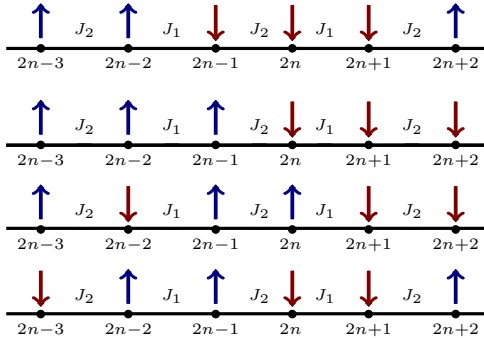

FIG. 4. Configurations for the constraint equations for three magnons. These states have restricted scattering configurations under exchange processes when compared to configurations that satisfy the hopping equations. Sites not displayed are in the all-up configuration.

to enumerate all these states and provide closed-form expressions using the GBA, where applicable.

We adopt the notation $|i\ j\ k\rangle \equiv S_i^- S_j^- S_k^- |F\rangle$ for $i < j < k$. Since the three magnons can be on the same or different sublattices, there are 8 categories of coefficients in the three-magnon wave function,

$$|\psi_{3mag}\rangle = \sum_{n<m<l} a_{n,m,l} |n, m, l\rangle. \qquad (38)$$

Analogous to what was done in previous sections, our task is to obtain the functional form of $a$'s that satisfy the Schrödinger equation $H|\psi_{3mag}\rangle = \varepsilon|\psi_{3mag}\rangle$.

Consider first the case where none of the three magnons are nearest neighbors, we get 8 categories of hopping equations which can be organized as 1 (all three magnons on $A$) + 3 (two magnons on $A$, one on $B$) + 3 (two magnons on $B$, one on $A$) + 1 (all three magnons on $B$). We define $E \equiv \varepsilon - (\Delta_1 + \Delta_2)(N-6)/4$ and explicitly write out only two representative equations, with the understanding that the other six are similar and straightforward to obtain,

$$E\, a_{2n,2m,2l} = \frac{J_1}{2}(a_{2n+1,2m,2l} + a_{2n,2m+1,2l} + a_{2n,2m,2l+1}) + \frac{J_2}{2}(a_{2n-1,2m,2l} + a_{2n,2m-1,2l} + a_{2n,2m,2l-1}), \tag{39a}$$

$$E\, a_{2n+1,2m,2l} = \frac{J_1}{2}(a_{2n,2m,2l} + a_{2n+1,2m+1,2l} + a_{2n+1,2m,2l+1}) + \frac{J_2}{2}(a_{2n+2,2m,2l} + a_{2n+1,2m-1,2l} + a_{2n+1,2m,2l-1}). \tag{39b}$$

There are two types of constrained configurations, representative situations have been shown in Fig. 4. First, when all three magnons are on consecutive sites, which corresponds to their locations being $(2n, 2n+1, 2n+2)$ or $(2n+1, 2n+2, 2n+3)$, we get the two constraint equations,

$$\left(E - \Delta_1 - \Delta_2\right)a_{2n,2n+1,2n+2} = \frac{J_1}{2}a_{2n,2n+1,2n+3} + \frac{J_2}{2}a_{2n-1,2n+1,2n+2}, \tag{40a}$$

$$\left(E - \Delta_1 - \Delta_2\right)a_{2n+1,2n+2,2n+3} = \frac{J_1}{2}a_{2n,2n+2,2n+3} + \frac{J_2}{2}a_{2n+1,2n+2,2n+4}. \tag{40b}$$

Next, an additional 8 constraint equations correspond to situations where exactly two magnons are nearest neighbors – these correspond to magnon locations of $(2n, 2m, 2m+1)$, $(2n, 2m+1, 2m+2)$, $(2n, 2n+1, 2m+2)$, $(2n, 2n+1, 2m+1)$, $(2n+1, 2m+2, 2m+3)$, $(2n+1, 2m+1, 2m+2)$, $(2n+1, 2n+2, 2m+2)$, $(2n+1, 2n+2, 2m+3)$ for $m > n$. We write out equations for four representative configurations,

$$\left(E - \Delta_1\right)a_{2n,2m,2m+1} = \frac{J_1}{2}a_{2n+1,2m,2m+1} + \frac{J_2}{2}\left(a_{2n-1,2m,2m+1} + a_{2n,2m-1,2m+1} + a_{2n,2m,2m+2}\right), \tag{41a}$$

$$\left(E - \Delta_1\right)a_{2n-1,2m,2m+1} = \frac{J_1}{2}a_{2n-2,2m,2m+1} + \frac{J_2}{2}\left(a_{2n,2m,2m+1} + a_{2n-1,2m-1,2m+1} + a_{2n-1,2m,2m+2}\right), \tag{41b}$$

$$\left(E - \Delta_2\right)a_{2n,2m+1,2m+2} = \frac{J_2}{2}a_{2n-1,2m+1,2m+2} + \frac{J_1}{2}\left(a_{2n+1,2m+1,2m+2} + a_{2n,2m,2m+2} + a_{2n,2m+1,2m+3}\right), \tag{41c}$$

$$\left(E - \Delta_2\right)a_{2n+1,2m+1,2m+2} = \frac{J_2}{2}a_{2n+2,2m+1,2m+2} + \frac{J_1}{2}\left(a_{2n,2m+1,2m+2} + a_{2n+1,2m,2m+2} + a_{2n+1,2m+1,2m+3}\right). \tag{41d}$$

Thus there are a total of 10 constraint equations. (Note that since the system and wave functions have translational symmetry, fewer constraint equations will need to be examined when attempting a GBA solution.)

In the rest of the section, we will show the application of the two-step GBA approach to the three-magnon problem. While we have presented the hopping and constraint equations for the general $XXZ$ case, we will focus on the AHC ($J_1 = \Delta_1 = -J_2 = -\Delta_2 = 1$) and zero energy states ($\varepsilon = E = 0$).

### A. Solutions of the hopping equations

Consider the solution of the hopping equations given by the product of three plane waves,

$$a_{n,m,l} = \alpha_{f(n),f(m),f(l)}e^{i(k_1 n + k_2 m + k_3 l)}, \tag{42}$$

with apriori unknown coefficients $\alpha_{f(n),f(m),f(l)}$ and momenta $k_1, k_2, k_3$. On plugging this form into the 8 hopping equations, we note that they organize themselves into two families of coefficients – four equations relate $\alpha_{AAA}, \alpha_{BBA}, \alpha_{ABB}, \alpha_{BAB}$ (odd number of $A$) and the other four relate $\alpha_{BBB}, \alpha_{AAB}, \alpha_{BAA}, \alpha_{ABA}$ (even num-

ber of $A$). These equations in matrix-vector form are,

$$\begin{pmatrix} +\sin k_1 & +\sin k_2 & +\sin k_3 & 0 \\ +\sin k_2 & +\sin k_1 & 0 & -\sin k_3 \\ 0 & +\sin k_3 & +\sin k_2 & -\sin k_1 \\ +\sin k_3 & 0 & +\sin k_1 & -\sin k_2 \end{pmatrix} \begin{pmatrix} \alpha_{BAA} \\ \alpha_{ABA} \\ \alpha_{AAB} \\ \alpha_{BBB} \end{pmatrix} = \begin{pmatrix} 0 \\ 0 \\ 0 \\ 0 \end{pmatrix} \tag{43}$$

and

$$\begin{pmatrix} +\sin k_1 & +\sin k_2 & +\sin k_3 & 0 \\ +\sin k_2 & +\sin k_1 & 0 & -\sin k_3 \\ 0 & +\sin k_3 & +\sin k_2 & -\sin k_1 \\ +\sin k_3 & 0 & +\sin k_1 & -\sin k_2 \end{pmatrix} \begin{pmatrix} \alpha_{ABB} \\ \alpha_{BAB} \\ \alpha_{BBA} \\ \alpha_{AAA} \end{pmatrix} = \begin{pmatrix} 0 \\ 0 \\ 0 \\ 0 \end{pmatrix}. \tag{44}$$

Note that the same $4 \times 4$ matrix appears in both sets of equations. A non-trivial solution exists if the determinant of this matrix is zero, which is equivalent to the condition that the sum of non-interacting energies of the three magnons is zero, i.e.

$$\sin k_1 \pm \sin k_2 \pm \sin k_3 = 0, \tag{45}$$

where our notation indicates that any one out of four conditions must be exactly satisfied.

The phase space of allowed solutions is now expectedly much larger than the two-magnon case. For example, some (not all) of the momentum combinations that

satisfy the energy conditions, up to permutations, are

$$k_1 = +k, \quad k_2 = \pm k, \quad k_3 = 0, \tag{46a}$$

$$k_1 = \pm k, \quad k_2 = +k + \frac{2\pi}{3}, \quad k_3 = +k - \frac{2\pi}{3}, \tag{46b}$$

where we have not constrained $k$ in any way yet. Note that the solutions $k_1 = k_2 + \pi$, $k_3 = \pi$ or $k_1 = \pm k$, $k_2 = k - \frac{\pi}{3}$, and $k_3 = k + \frac{\pi}{3}$ are equivalent to the solutions above, since in the case of the two-magnon solutions any phase factors generated by additional $\pi$'s can be absorbed into the definition of the Bethe coefficients.

## B. Generalized Bethe ansatz solutions for zero total momentum for arbitrary $N$

Consider GBA states which are linear combinations of $(k, -k, 0)$ and $(0, 0, 0)$. For general $k \neq 0$, six permutations of the momentum labels are possible. The GBA for the 8 families of three-magnon coefficients is

characterized by 56 parameters (8 families $\times$ 6 permutations $+$ 8 constants), which is cumbersome to deal with analytically. To reduce the number of parameters, we take inspiration from the two-magnon solutions in Sec. III. We classify the net zero momentum solutions as having eigenvalue $R = 1$ or $R = -1$ under $\tau$. Additionally, we impose periodic boundary conditions on the three-magnon wave function coefficients, for $n < m < l$, $a_{n,m,l} = a_{m,l,n+2N} = a_{l,n+2N,m+2N}$.

To keep our notation brief for each class of GBA, we write out expressions for only 4 of the 8 families of coefficients, because the other 4 families are related by symmetry. For example, for $R = 1$, $a_{2n,2m,2l} = a_{2n+1,2m+1,2l+1}$ and $a_{2n,2m,2l+1} = a_{2n+1,2m+1,2l+2}$. For $R = -1$, $a_{2n,2m,2l} = -a_{2n+1,2m+1,2l+1}$ and $a_{2n,2m,2l+1} = -a_{2n+1,2m+1,2l+2}$. In what follows we also use the same labels $\alpha, \beta, \gamma, \delta, C_1, C_2$ for the unknown parameters for either family, but it is understood that, in general, they assume different values.

For the $R = 1$ family, we get,

$$a_{2n,2m,2l} = \alpha e^{ik(2n-2m)} + \beta e^{ik(2m-2n)} + \alpha e^{ik(2m-2l)} + \beta e^{ik(2l-2m)} + \alpha e^{-2ikN} e^{ik(2l-2n)} + \beta e^{ik(2n-2l+2N)} + C_1, \tag{47a}$$

$$a_{2n,2m,2l+1} = -\alpha e^{ik(2n-2m)} - \beta e^{ik(2m-2n)} + \gamma e^{ik(2m-2l-1)} + \delta e^{ik(2l-2m+1)} + \gamma e^{ik(2l-2n+1-2N)} + \delta e^{ik(2n-2l-1+2N)} + C_2, \tag{47b}$$

$$a_{2n,2m+1,2l} = \gamma e^{ik(2n-2m-1)} + \delta e^{ik(2m-2n+1)} + \gamma e^{ik(2m-2l+1)} + \delta e^{ik(2l-2m-1)} - \alpha e^{ik(2l-2n-2N)} - \beta e^{ik(2n-2l+2N)} + C_2, \tag{47c}$$

$$a_{2n+1,2m,2l} = \gamma e^{ik(2n-2m+1)} + \delta e^{ik(2m-2n-1)} - \alpha e^{ik(2m-2l)} - \beta e^{ik(2l-2m)} + \gamma e^{ik(2l-2n-1-2N)} + \delta e^{ik(2n-2l+1+2N)} + C_2, \tag{47d}$$

and for the $R = -1$ family we get,

$$a_{2n,2m,2l} = \alpha e^{ik(2n-2m)} + \beta e^{ik(2m-2n)} + \alpha e^{ik(2m-2l)} + \beta e^{ik(2l-2m)} + \alpha e^{ik(2l-2n-2N)} + \beta e^{ik(2n-2l+2N)} + C_1, \tag{48a}$$

$$a_{2n,2m,2l+1} = \alpha e^{ik(2n-2m)} + \beta e^{ik(2m-2n)} + \gamma e^{ik(2m-2l-1)} + \delta e^{ik(2l-2m+1)} + \gamma e^{ik(2l-2n+1-2N)} + \delta e^{ik(2n-2l-1+2N)} + C_2, \tag{48b}$$

$$a_{2n,2m+1,2l} = \gamma e^{ik(2n-2m-1)} + \delta e^{ik(2m-2n+1)} + \gamma e^{ik(2m-2l+1)} + \delta e^{ik(2l-2m-1)} + \alpha e^{ik(2l-2n-2N)} + \beta e^{ik(2n-2l+2N)} + C_2, \tag{48c}$$

$$a_{2n+1,2m,2l} = \gamma e^{ik(2n-2m+1)} + \delta e^{ik(2m-2n-1)} + \alpha e^{ik(2m-2l)} + \beta e^{ik(2l-2m)} + \gamma e^{ik(2l-2n-1-2N)} + \delta e^{ik(2n-2l+1+2N)} + C_2. \tag{48d}$$

Thus use of symmetry leaves us with the problem of determining far fewer parameters, only six of them, and we will soon see we do not need all of them. We re-emphasize that the GBA parameterizations of the three-magnon states satisfy the hopping equations by construction, the task is to determine $\alpha, \beta, \gamma, \delta, C_1, C_2$ that satisfy the constraint equations. The use of symmetry also reduces the number of constraint equations to look at especially when we work in the net zero momentum sector. Additionally, in the results that follow we use symmetry to compactly write out the three-magnon eigensolutions, specifically, analytic expressions for $a_{2n,2m,2l}$ and $a_{2n,2m,2l+1}$ exactly determine the expressions of the other six families of coefficients and thus are not explicitly written out.

### 1. Generalized Bethe ansatz zero energy states for $R = 1$

Consider the constraint equation for configurations where the three magnons exist on contiguous sites $2n, 2n + 1, 2n + 2$,

$$a_{2n,2n+1,2n+3} - a_{2n-1,2n+1,2n+2} = 0. \tag{49}$$

Plugging in the symmetry adapted GBA from Eq. (47) into Eq.(49) we find that it is satisfied for *arbitrary* $\alpha, \beta, \gamma, \delta, C_1, C_2$. So this equation does not constrain the Bethe parameters in any way. Symmetry guarantees that the same conclusion holds for the constraint equation for three magnons on sites $2n + 1, 2n + 2, 2n + 3$.

Consider next the second constraint equation where two magnons are nearest neighbors, but the third is not – take the three magnons to be located at sites $2n, 2m +$

1, $2m + 2$ with $m > n$. The constraint equation for this case reads

$$2a_{2n,2m+1,2m+2} = a_{2n,2m,2m+2} + a_{2n,2m+1,2m+3} \qquad (50)$$
$$+ a_{2n+1,2m+1,2m+2} - a_{2n-1,2m+1,2m+2}.$$

We use the GBA from Eq. (47) in Eq. (51). Carrying out some algebra (not shown) and choosing $k = p\pi/N$ where $p \in \mathbb{Z}$, i.e. $e^{2ikN} = 1$, we find

$$e^{ik(2n-2m-1)}\left(\gamma - \delta - 3\beta e^{-ik} - \alpha e^{-ik}\right)$$
$$+ e^{ik(2m+1-2n)}\left(\delta - \gamma - 3\alpha e^{ik} - \beta e^{ik}\right)$$
$$+ \left(2\gamma e^{-ik} + 2\delta e^{ik} + C_2 - C_1\right) = 0. \qquad (51)$$

This implies that the following three equations must hold,

$$(\gamma - \delta) - (3\beta + \alpha)e^{-ik} = 0, \qquad (52a)$$
$$(\gamma - \delta) + (3\alpha + \beta)e^{ik} = 0, \qquad (52b)$$
$$2\gamma e^{-ik} + 2\delta e^{ik} + C_2 - C_1 = 0. \qquad (52c)$$

Consider the case of $\alpha = \beta = 0$. The first two equations are satisfied when $\gamma = \delta$. Then plugging this into the third equation gives $C_2 - C_1 = -4\gamma \cos k$. Since only $C_2 - C_1$ matters we can choose $C_1 = 0$ and $C_2 = -4\gamma \cos k$. After choosing an arbitrary scale factor

$\gamma = 1/2$, we arrive at the solution,

$$\alpha = \beta = 0, \qquad \gamma = \delta = +1/2, \qquad (53a)$$
$$C_1 = 0, \qquad C_2 = -2\cos k. \qquad (53b)$$

which corresponds to the three-magnon wave function,

$$a_{2n,2m,2l} = 0, \qquad (54a)$$
$$a_{2n,2m,2l+1} = \cos(k(2m - 2l - 1))$$
$$+ \cos(k(2n - 2l - 1)) - 2\cos k, \qquad (54b)$$

where we have written expressions for only two out of eight families of coefficients, with the understanding that the use of symmetry (periodic boundary conditions and $R = 1$) can be used to arrive at expressions for the remaining coefficients. We observe that this state is just the spin-lowered descendant of the $R = 1$ two-magnon $AB$ solution. While this is completely expected in accordance with SU(2) symmetry, it is reassuring that it emerges from our solution strategy. Note that, just like the case of two magnons, $k = 0$ and $k = \pi/2$ lead to wave functions that vanish and are hence not physical.

Interestingly, the set of three equations in Eq. (52) admit other linearly independent solutions i.e. those that are not descendants of two-magnon zero modes. To see this, we now do not impose $\alpha = \beta = 0$. Choosing $\gamma = -\delta = \delta^* = i/2$ and $C_2 = 0$ we get,

$$\alpha = -i\left(\frac{3e^{-ik} + e^{ik}}{8}\right) = \beta^*, \qquad (55a)$$
$$\gamma = +i/2 = \delta^*, \qquad (55b)$$
$$C_1 = +2\sin k, \qquad C_2 = 0. \qquad (55c)$$

These parameters yield the three-magnon wave function

$$a_{2n,2m,2l} = \frac{3}{4}\left(\sin(k(2n - 2m - 1)) + \sin(k(2m - 2l - 1)) + \sin(k(2l - 2n - 1))\right)$$
$$+ \frac{1}{4}\left(\sin(k(2n - 2m + 1)) + \sin(k(2m - 2l + 1)) + \sin(k(2l - 2n + 1))\right) + 2\sin k, \qquad (56a)$$
$$a_{2n,2m,2l+1} = -\frac{3}{4}\sin(k(2n - 2m - 1)) - \frac{1}{4}\sin(k(2n - 2m + 1)) - \sin(k(2m - 2l - 1)) - \sin(k(2l - 2n + 1)). \qquad (56b)$$

Observe that the case $k = 0$ is unphysical.

We also remark that the third class of constraint equation (for magnons at sites $2n, 2m, 2m + 1$ with $m > n$) was not explicitly presented here, but it yields the same equations for $\alpha, \beta, \gamma, \delta, C_1, C_2$, and thus the solutions we have written in Eq. (54) and Eq. (56) still hold.

2. *Generalized Bethe ansatz zero energy states for $R = -1$*

The arithmetic for the $R = -1$ case has many similarities to the $R = 1$ case, which we refer to when discussing the solutions. For example, the constraint equation corresponding to three magnons on consecutive sites is once again satisfied for *arbitrary $\alpha, \beta, \gamma, \delta, C_1, C_2$*.

Consider magnons at sites $2n, 2m + 1, 2m + 2$ with $m > n$. The constraint equation for this case yields,

for $e^{2ikN} = 1$,

$$e^{ik(2n-2m-1)}\Big(3\gamma + \delta + \beta e^{-ik} - \alpha e^{-ik}\Big)$$
$$+ e^{ik(2m-2n+1)}\Big(3\delta + \gamma + \alpha e^{ik} - \beta e^{ik}\Big)$$
$$+ \Big(2\gamma e^{-ik} + 2\delta e^{ik} + C_2 - C_1\Big) = 0. \qquad (57)$$

This means that the following three equations must hold,

$$(\beta - \alpha) - (3\delta + \gamma)e^{-ik} = 0, \qquad (58a)$$
$$(\beta - \alpha) + (3\gamma + \delta)e^{ik} = 0, \qquad (58b)$$
$$2\gamma e^{-ik} + 2\delta e^{ik} + C_2 - C_1 = 0. \qquad (58c)$$

The first two equations look similar to that of the $R = 1$ case, except with the roles of $\gamma, \delta$ and $\beta, \alpha$ exchanged. The equation governing the constants is identical in both systems. For $e^{2ikN} = 1$, we find one of the solution sets (using an arbitrary scale factor $\alpha = 1/2$ and the choice of constant to be both zero) to be,

$$C_1 = C_2 = \gamma = \delta = 0, \qquad (59a)$$
$$\alpha = \beta = +1/2, \qquad (59b)$$

which corresponds to the three-magnon wave function

$$a_{2n,2m,2l} = \cos(k(2n - 2m)) + \cos(k(2m - 2l))$$
$$+ \cos(k(2l - 2n)), \qquad (60a)$$
$$a_{2n,2m,2l+1} = \cos(k(2n - 2m)). \qquad (60b)$$

This is the descendant state obtained by acting with the total spin-lowering operator on the $R = -1$ two-magnon state.

To obtain the second class of $R = -1$ solutions we recognize the parallels with the $R = 1$ family and solve for the constants (choosing $C_2 = 0$). We find,

$$\gamma = -i\Big(\frac{3e^{-ik} + e^{ik}}{8}\Big) = \delta^*, \qquad (61a)$$
$$\alpha = -i/2 = \beta^*, \qquad (61b)$$
$$C_1 = -\frac{3}{2}\sin 2k, \qquad C_2 = 0, \qquad (61c)$$

which corresponds to the three-magnon wave function,

$$a_{2n,2m,2l} = \sin(k(2n - 2m)) + \sin(k(2m - 2l))$$
$$+ \sin(k(2l - 2n)) - \frac{3}{2}\sin 2k, \qquad (62a)$$
$$a_{2n,2m,2l+1} = \sin(k(2n - 2m))$$
$$+ \frac{3}{4}\Big(\sin(k(2m - 2l - 2)) + \sin(k(2l - 2n))\Big)$$
$$+ \frac{1}{4}\Big(\sin(k(2m - 2l)) + \sin(k(2l - 2n + 2))\Big). \qquad (62b)$$

Clearly $k = 0$ and $k = \frac{\pi}{2}$ yield wave functions that vanish everywhere and hence must not be considered.

### 3. Counting the total number of $R = \pm 1$ zero modes

The arguments for counting the number of linearly independent three-magnon states parallel those we encountered earlier in the two-magnon case. For both even and odd $N$, there are $N$ two-magnon states in the zero momentum sector (including the uniform mode) and hence $N$ three-magnon states that descend from these on lowering spin. For even $N$, there are additional $\frac{N}{2} - 1$ linearly independent solutions (each) in the $R = 1$ and $R = -1$ sectors. (We find that the uniform mode is not linearly independent of these additional $R = 1$ modes and this is accounted for in the count). This brings the total number of states for even $N$ (all of them in the zero momentum sector) to $N + 2(\frac{N}{2} - 1) = 2(N - 1)$, thus explaining the counts seen in Table I.

For odd $N$, there are $\frac{N-1}{2} - 1$ linearly independent modes of the $R = 1$ type and $\frac{N-1}{2}$ modes of the $R = -1$ type which are not descendants of the two-magnon zero modes. The overall number (zero momentum descendants and non-descendants) of $R = 1$ modes and $R = -1$ modes, however, are both equal to $(N - 1)$. Thus the total number of linearly independent zero modes in the zero momentum sector is $2(N - 1)$; the formula is identical to the case of even $N$. However, Table I shows that the total number (across all sectors) for odd $N$ is $4(N - 1)$; we address this discrepancy next.

### C. Additional solutions for odd $N$

Much like the two-magnon case, for odd $N$, we numerically find that the exact additional zero energy solutions have net non-zero momentum $K$. To attempt a GBA solution, we first identify momentum combinations that satisfy the conditions (1) $k_1 + k_2 + k_3 = K$ (modulo $\pi$), and (2) any one of the four equations $\sin k_1 \pm \sin k_2 \pm \sin k_3 = 0$. Not worrying about permutations of the momenta, we need to consider only two of these four cases. The first case corresponds to,

$$\sin k_1 + \sin k_2 + \sin(K - k_1 - k_2) = 0. \qquad (63)$$

We use the fact that this is a quadratic equation in the variable $e^{ik_2}$ which can be readily solved to give,

$$e^{ik_2} = \frac{-i\sin k_1 \pm \sqrt{4\sin^2\frac{K-k_1}{2} - \sin^2 k_1}}{1 - e^{-i(K-k_1)}}. \qquad (64)$$

The second case corresponds to

$$\sin k_1 - \sin k_2 + \sin(K - k_1 - k_2) = 0, \qquad (65)$$

which gives,

$$e^{ik_2} = \frac{i\sin k_1 \pm \sqrt{4\cos^2\frac{K-k_1}{2} - \sin^2 k_1}}{1 + e^{-i(K-k_1)}}. \qquad (66)$$

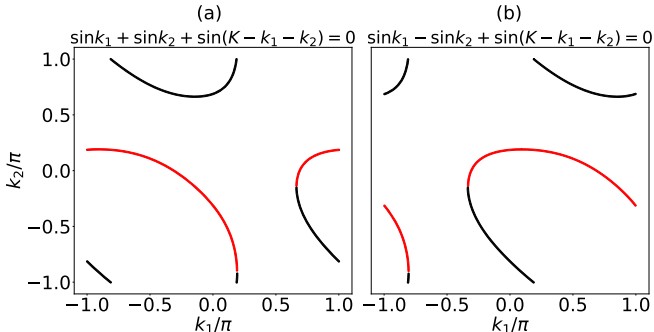

FIG. 5. Representative example of points in the $(k_1, k_2)$ plane that satisfy the zero energy condition for $K = 2\pi n/N$ where $n = 3$ and $N = 16$. Panel (a) corresponds to Eq. (63) and (b) corresponds to Eq. (65). In each case, the red color corresponds to the "+" sign between the terms and the black color corresponds to the "−" sign in Eqns. (64) and (66).

| $k_1$ | $k_2$ | $k_3$ |
|---|---|---|
| $-K$ | $K + \frac{2\pi}{3}$ | $K - \frac{2\pi}{3}$ |
| $\frac{K}{2}$ | $\frac{K}{2}$ | $0$ |
| $\frac{K}{2} + \frac{\pi}{2}$ | $\frac{K}{2} - \frac{\pi}{2} \equiv \frac{K}{2} + \frac{\pi}{2}$ | $0$ |
| $\frac{K}{3}$ | $\frac{K}{3} + \frac{2\pi}{3}$ | $\frac{K}{3} - \frac{2\pi}{3}$ |

TABLE II. Momentum sets with total momentum $K$, that satisfy the hopping equations and the zero total energy constraint, used in the three-magnon numerical generalized Bethe ansatz calculation. Permutations of the momenta are not shown.

In Fig. 5 we show a representative example of points in the $(k_1, k_2)$ plane that satisfies the zero energy condition, for $K = 2\pi n/N$ where $n = 3$ and $N = 16$. In Table II we also show some special solutions (up to permutations, and up to individual phase shifts of $\pi$) that satisfy either Eq. (63) or Eq. (65). We attempt GBA solutions only for momentum combinations in Table II.

### 1. Linear combination of Bethe states with momenta $k_1 = k_2 = \frac{K}{2}$, $k_3 = 0$ and $k_1 = k_2 = \frac{K}{2} + \frac{\pi}{2}$ and $k_3 = 0$

We first attempt a linear combination for a subset of momenta in Table II, in particular, we consider only $(k, k, 0)$ and $(k + \frac{\pi}{2}, k + \frac{\pi}{2}, 0)$. The only zero energy state we find is the spin-lowered descendant of the two-magnon state discussed in Sec. III D. In Appendix C we provide a complete proof for why this parameterization yields only a single solution for a given $K$, and for completeness, we also write out the expression for this family of states.

It follows that the number of linearly independent states in this family is $N - 1$. When combined with the number of states in the momentum zero sector, all of which are described by the GBA, the total number obtained is $3(N-1)$. Thus, we are left with $N-1$ states, one

in each non-zero momentum sector, still to be explained.

### 2. Numerical GBA search and possible non-Bethe solutions

For a given $K = \frac{2\pi n}{N}$ where $n$ is a non-zero integer, we now utilize Bethe basis functions for *all* four momentum combinations in Table II along with their permutations. Note that the combination of sublattices, i.e. $AAA$, $AAB$, $ABA$, ..., $BBB$, and the momenta $k_1, k_2, k_3$ are needed to specify a single Bethe basis vector. Said differently, each momentum combination produces 8 different basis functions and, in general, there are 6 permutations of the momenta themselves (fewer if two or more momenta are equal to one another). Since this leads to a large number of free parameters, the analytic equations become somewhat formidable. Instead, we numerically diagonalize the square of Hamiltonian matrix in the basis of the Bethe vectors specified above. This basis is appropriately modified to eliminate linearly dependent vectors and explicitly orthonormalized (using numerical Gram Schmidt or related techniques). We refer to this diagonalization procedure as the "numerical generalized Bethe ansatz" (NGBA). The procedure requires the basis functions (the Bethe momenta) as the only input.

To see why it is important to diagonalize $H^2$ (and not just $H$) in the restricted Bethe basis, consider a state normalized $|\psi\rangle$ in this basis that satisfies

$$H|\psi\rangle = \alpha|\phi\rangle \tag{67}$$

where $\alpha$ is a scalar and where $|\phi\rangle$ (assumed normalized) is not in the span of the Bethe basis. By definition $\langle\psi|\phi\rangle = 0$. Then diagonalization of $H$ in the Bethe basis would suggest that the energy $\langle\psi|H|\psi\rangle = 0$, however, the state is clearly not an exact eigenstate of $H$ unless $\alpha$ is strictly zero. Given that $H$ is Hermitian we have,

$$H|\phi\rangle = \alpha^*|\psi\rangle + \beta|\phi\rangle + \gamma|\phi'\rangle \tag{68}$$

where $|\phi'\rangle$ satisfies $\langle\phi|\phi'\rangle = \langle\psi|\phi'\rangle = 0$ and where $\beta$ and $\gamma$ are scalars. Thus,

$$\langle\psi|H^2|\psi\rangle = |\alpha|^2 \tag{69}$$

which implies that diagonalizing $H^2$ in the Bethe basis is a sure test of an exact zero energy eigenstate. This is not unexpected, after all, the variance of the energy $\langle\psi|H^2|\psi\rangle - (\langle\psi|H|\psi\rangle)^2$ must be zero if and only if $|\psi\rangle$ is an exact eigenstate of $H$. We remark that even though zero energy states are the focus of our current investigation, the procedure outlined is perfectly general and could be applied to search for exact GBA eigenstates at any energy, if at all they exist in the model of interest.

Our NGBA calculation for three magnons reveals that while the Bethe basis is sufficiently flexible for describing more than one zero energy state for $N = 3$ and $N = 5$ (for a given fixed $n \neq 0$), it captures only one exact zero energy state for $N > 5$. This zero energy state is just a

descendant of a two-magnon state with the momentum combination discussed above in Sec. IV C 1. While we can not rule out that other simple parameterizations may account for the missing $N-1$ states, the restricted Bethe basis is insufficient for this purpose and thus we classify the state as a "non-Bethe solution".

## V. PAIRED MAGNONS FROM THE NUMERICAL GENERALIZED BETHE ANSATZ

We now discuss the nature of the GBA eigenfunctions of multi-magnon states at zero energy, more specifically, we investigate whether pairs of magnons have opposite momenta $k, -k$ (i.e. they "pair up"). The smallest number of magnons required to test this idea is four - there are two pairs of magnons - and we explore this with the NGBA. We focus on even $N$, and eigenstates with zero net momentum - the largest number of zero energy states exist in this symmetry sector.

We present a concrete example of a sequence of NGBA calculations that we carried out for $N = 8$, with the aim of finding one or more generalized Bethe eigenfunctions which take two fixed $k_1 = \frac{2\pi n_1}{2N}$ and $k_2 = \frac{2\pi n_2}{2N}$ where $n_1$ and $n_2$ are integers, that produces a total zero momentum, zero total energy exact eigenstate. The sequence of NGBA calculations and the corresponding results are summarized in Table III where we set $n_1 = 2$ and $n_2 = 3$.

We first attempt a GBA solution using only basis functions with momenta $(k_1, -k_1, k_2, -k_2)$ and their permutations. (Note that for most generic distinct momenta there are $4! = 24$ permutations, exceptions arise when either $k_1$ or $k_2$ or both are zero or if $k_1 = k_2$ or $k_1 = -k_2$. From here on the incorporation of permuted basis functions will be assumed and not stated explicitly). For our example ($k_1 \neq 0$ and $k_2 \neq 0$), we found that these basis functions alone did not yield any exact zero energy state. Our experience with two- and three-magnon wave functions suggests that the next natural set of basis functions to incorporate are those with momentum $(0, 0, 0, 0)$. When treated in isolation, the $(0, 0, 0, 0)$ basis must yield exactly two zero energy states (1) The uniform mode $(S_{\text{tot}}^-)^4 |F\rangle$ where $S_{\text{tot}}^- = S_{k=0,A}^- + S_{k=0,B}^-$, which is a descendant of the ferromagnetic state and (2) the direct descendant of the "staggered" one-magnon zero energy state $(S_{\text{tot}}^-)^3 (S_{k=0,A}^- - S_{k=0,B}^-) |F\rangle$. Our NGBA calculations confirm this result.

Next we combine basis functions from momenta $(0, 0, 0, 0)$ and $(k_1, -k_1, k_2, -k_2)$. We find no additional zero energy states, suggesting the need for more basis functions to produce a GBA eigenstate. We thus appeal to basis functions with momenta $(k_1, -k_1, 0, 0)$ and $(k_2, -k_2, 0, 0)$. When these two sets of momenta are combined with momentum $(0, 0, 0, 0)$ a total of 10 zero energy states are found. 8 of these zero energy states are descendants of the three-magnon states: for each $k_i$ there are two $R = 1$ and two $R = -1$ states (see Sec. IV B). The two states made purely from $(0,0,0,0)$ also carry over.

When all basis functions are incorporated, the NGBA produces a total of 15 zero energy states. This implies that the incorporation of $(k_1, -k_1, k_2, -k_2)$ led to $(15 - 10) = 5$ additional zero energy states that are genuinely new four magnon Bethe eigenstates with no direct symmetry relation with the lower-magnon states. This is reminiscent of our findings on going from two to three magnons - in addition to the spin-lowered descendant states new Bethe solutions were found analytically.

While $k_1$ and $k_2$ have been fixed to specific values in the example above, we note that they could assume other values, including fractional momenta. Here we restrict our NGBA analysis to lattice momenta $k_i = 2\pi n_i / 2N$. We perform one NGBA calculation for every $(n_1, n_2)$ with $0 < n_1 \leq n_2 \leq N/2$ (note: $n_1 \leq 0$ and $n_2 \leq 0$ are already accounted for in the choice of basis functions). We then collect the exact zero energy eigenstates from each individual NGBA calculation and determine the rank of this collective space. The number of linearly independent states obtained across all $n_1, n_2$ is the total number of four-magnon zero energy Bethe states with a compact GBA representation, and we call the corresponding method "sequential NGBA". We have also carried out the "one-step" single NGBA calculation where all basis functions incorporating all $(n_1, n_2)$ pairs were introduced in one step, this allows for more general GBA to be admissible. We find this flexibility to be important for the explanation of zero energy eigenstates that are not descendants of three-magnon states for $N \geq 10$. Note that the one-step procedure is equivalent to choosing only a fraction of basis states that satisfy $k_1 + k_2 + k_3 + k_4 = 0$ and one of the conditions $\sin k_1 \pm \sin k_2 \pm \sin k_3 \pm \sin k_4 = 0$; it is a diagonalization in a restricted subspace.

Table IV shows the results of our "sequential" and "one-step" NGBA calculations compared to exact diagonalization, in the zero total momentum sector. For $N = 4, 6$ both capture all four-magnon zero energy states, and for $N = 8$ they capture a majority of the 4-magnon states (24 out of 28). For larger $N$, however, our sequential NGBA procedure captures only descendants of the three-magnon states. The increased flexibility that the one-step NGBA offers is advantageous in these cases and we observe that additional modes are now indeed captured. However, there are many four-magnon zero modes that we have not been able to explain with our choice of basis functions and we have not been able to determine why this is the case – does it imply that these additional states do not have a GBA description with paired magnons or, alternatively, is the choice of lattice momenta insufficient?

We conclude this section by remarking on some observations. Table IV shows that the exact number of 4-magnon states for even $N$ is $\binom{N}{2}$ and Table I shows that the number of six-magnon zero energy states across all momentum sectors (but dominated by $K = 0$) is close to (or exactly) $\binom{N}{3}$. Such counts would be expected to be associated with two or three "pairs" of momenta. Assuming this combinatorial structure persists for an even

| Momentum sets in NGBA | Basis size | # zero energy states | Reason |
|---|---|---|---|
| $(k_1, -k_1, k_2, -k_2)$ only | 384 | 0 | Basis too constrained |
| $(0,0,0,0)$ only | 16 | 2 | Descendants of the uniform and staggered one magnon zero energy states |
| $(0,0,0,0)+ (k_1, -k_1, k_2, -k_2)$ | 400 | 2 | zero energy states come only from $(0,0,0,0)$ |
| $(0,0,0,0) + (k_1, -k_1, 0, 0)$, $(k_2, -k_2, 0, 0)$ | 385 | 10 | 8 descendants of three-magnon states i.e. 4 states for each $(k_i, -k_i, 0, 0)$ combined with $(0,0,0,0)$ (2 each in $R = \pm 1$ sectors) + 2 states from solely $(0,0,0,0)$ |
| All four-momentum sets | 689 | 15 | 5 states not direct descendants of 3-magnon states |
| All + $(k_1, -k_1, k_1, -k_1)$ + $(k_2, -k_2, k_2, -k_2)$ | 695 | 17 | 7 states not direct descendants of 3-magnon states |

TABLE III. Results of NGBA calculations for four magnons in a system of $N = 8$ unit cells ($L = 16$ sites) for $k_1 = \frac{2\pi n_1}{2N}$ and $k_2 = \frac{2\pi n_2}{2N}$, for $n_1 = 2$ and $n_2 = 3$, for different basis sets. Permutations of the momenta are not shown. The "basis size" consists of only linearly independent basis functions.

| | Number of 4-magnon zero energy states | | | |
|---|---|---|---|---|
| $N$ | Exact ($K = 0$) | Sequential | One-step | Basis size |
| 4 | 6 | 6 | 6 | 55 |
| 6 | 15 | 15 | 15 | 284 |
| 8 | 28 | 24 | 24 | 764 |
| 10 | 45 | 18 | 30 | 1500 |
| 12 | 66 | 22 | 34 | 2492 |
| 14 | 91 | 26 | 37 | 3740 |
| 16 | 120 | 30 | 41 | 5244 |

TABLE IV. Results of NGBA calculations for four magnons for various even $N$, compared to exact diagonalization in the total momentum $K = 0$ sector. The "basis size" is the number of linearly independent generalized plane waves used in the one-step NGBA method. The details of the sequential and one-step NGBA methods are discussed in the text. Only states that have an energy of zero to within machine precision are considered in the counts.

higher number of magnons, it is conceivable that such a structure is at the heart of the explanation of the number of zero energy states which scales as $\approx 2^N$, seen by us in complementary work [51]. We do not have a definitive answer on this issue and leave its resolution to future work. (We do note that in a recent article some of us used a different methodology to construct exact volume-law-entangled states for the AHC [77]. Although this does not capture all the few-magnon solutions discussed here, it suggests that some non-trivial partial integrability can exist even within the higher-magnon sector.)

## VI. ENTANGLEMENT ENTROPY OF ZERO-ENERGY EIGENSTATES

In this section, we calculate the von Neumann entanglement entropy ($S_l^{\text{vN}}$) of the two- and three-magnon zero energy eigenstates and show that they are all area-law entangled states (their entanglement saturates to a constant as the system size is increased).

We first outline the general method of calculating $S_l^{\text{vN}}$ of an eigenstate $|\psi\rangle$ with a fixed number of magnons. Here $l$ is the subsystem size and we trace out the rest of the system (the environment of size $L - l$) to calculate the reduced density matrix of the subsystem, $\rho_l$. The Hamiltonian is block-diagonal in magnon number basis because of the global $U(1)$ symmetry and consequently, the eigenstates can be chosen as states with fixed magnon numbers. The reduced density matrix (RDM) is also block-diagonal in the magnon number basis of the subsystem, though the magnon number is not conserved in the subsystem. Hence, the structure of $\rho_l$ is as follows,

$$\rho_l = \rho_{l,0} \oplus \rho_{l,1} \oplus \rho_{l,2}, \tag{70}$$

where $\rho_{l,q}$ is the block with $q$ magnons in the subsystem of length $l$. These blocks are of size $\binom{l}{q}$ and we will construct and diagonalize each separately.

### A. Two-magnon states

In this section, we will calculate the entanglement entropy of the zero energy eigenstates with two magnons from Eq. (11). We will only consider the linearly independent momentum including $k = \pi/2$. The normalization of the state is now important to get the correct value of $S_l^{\text{vN}}$. Let us first consider the $AA - BB$ solutions [Eqs. (21)], for which the normalization constant $\mathcal{N}$ is given by,

$$\mathcal{N} = 2 \sum_{n \leq m} \cos^2(k(2n - 2m) \tag{71}$$

$$= \frac{1}{16} \csc^2(2k) \left( (L - 2)^2 - L(L - 4) \cos(4k) - 4 \right).$$

Next, we get (assuming the subsystem size $l$ to be even for simplicity),

$$\rho_{l,0} = \frac{1}{\mathcal{N}} \sum_{j=1}^{L-l-2} \sum_{\substack{i=2 \\ i \text{ even}}}^{L-l-j} a_{l+j,l+j+i}^2 \tag{72}$$

$$= \frac{1}{16\mathcal{N}} \csc^2(2k) \Big[ (L-l-2)^2$$
$$- (L-l)(L-l-4)\cos(4k) - 4\cos(2k(L-l)) \Big]$$
$$= \frac{(L-l-2)^2 - (L-l)(L-l-4)\cos(4k) - 4\cos(2k(L-l))}{(L-2)^2 - L(L-4)\cos(4k) - 4}.$$

Note that $\rho_{0,0} = 1$, as expected.

We now move on to calculate $\rho_{l,1}$ which is a $l \times l$ real hermitian matrix. The diagonal elements are given by,

$$\rho_{l,1}(x,x) = \frac{1}{\mathcal{N}} \sum_{i>l} a_{x,i}^2$$
$$= \frac{1}{4\mathcal{N}} \Big[ \csc(2k)\sin(2k(x-l-c))$$
$$+ \csc(2k)\sin(2k(L-x+c)) - l + L \Big], \tag{73}$$

where $c = 0(1)$ for $x$ odd (even). The off-diagonal elements are given by,

$$\rho_{l,1}(x,y) = \sum_{i>l} a_{x,i} a_{y,i} \tag{74}$$
$$= \frac{1}{2\mathcal{N}} \Big[ \frac{L-l}{2} \cos(k(x-y))$$
$$+ \csc(2k)\sin(k(L-l))\cos(k(l+L-x-y+c)) \Big],$$

where $c = 0(2)$ when $x$ and $y$ are both odd (even). $\rho_{l,1}(x,y) = 0$ when $x + y$ is odd. Though we can calculate the entire matrix analytically, diagonalization of $\rho_{l,1}$ is difficult in general and numerics is the only way forward. Interestingly, significant simplification can be achieved in some special cases, for example, most often we are interested in half-chain ($l = N$) entanglement entropy for which exact analytical expression of the spectrum of $\rho_{N,1}$ can be obtained. In this special case, the diagonal elements are given by,

$$\rho_{N,1}(x,x) = \begin{cases} \dfrac{1}{2N-2} & \text{for } k = 0, \pi/2, \\ \dfrac{N}{4\mathcal{N}} & \text{otherwise,} \end{cases} \tag{75}$$

and the nonzero off-diagonal elements ($x + y$ even) are given by,

$$\rho_{N,1}(x,y) = \begin{cases} \dfrac{\cos(2k)}{2N-2} & \text{for } k = 0, \pi/2, \\ \dfrac{N}{4\mathcal{N}} \cos(k(x-y)) & \text{otherwise.} \end{cases} \tag{76}$$

Note that the diagonal elements are all the same and the off-diagonal elements also have a simple structure.

It turns out that $\rho_{N,1}$ has only two nonzero eigenvalues $N/(4N-4)$ for the special momentum $k = 0, \pi/2$ and four nonzero eigenvalues $4N^2/(16\mathcal{N})$ for any other momentum. These eigenvalues are all degenerate.

We now calculate $\rho_{l,2}$. In this case, since no magnon resides in the environment, the partial trace does not involve any summation. Therefore $\rho_{l,2}$ is a rank-1 matrix and the corresponding eigenvalue is equal to its trace,

$$\text{Tr}[\rho_{l,2}] = \frac{1}{\mathcal{N}} \sum_{\substack{i+j \text{ even} \\ i<j\leq l}} \cos^2(k(i-j)) \tag{77}$$
$$= \frac{\csc^2(2k)}{16\mathcal{N}} ((l-2)^2 - l(l-4)\cos 4k - 4\cos 2lk).$$

Note that $\rho_{N,0} = \text{Tr}[\rho_{N,2}]$, since the subsystem and the environment are just exchanged in the two cases.

Now, since all the eigenvalues of the reduced density matrix are known, let us calculate the entanglement entropy. The half-chain entanglement entropy ($S_N^{\text{vN}}$), in the thermodynamic limit, is given by,

$$\lim_{\substack{N\to\infty \\ k=0,\pi/2}} S_N^{\text{vN}} = S_1\left( \frac{N}{4(N-1)}, \frac{N}{4(N-1)}, \frac{N-2}{4(N-1)}, \frac{N-2}{4(N-1)} \right)$$
$$= 2\ln 2, \tag{78}$$

in the cases of $k = 0, \pi/2$ where $S_1$ is the Shannon entropy of a discrete probability distribution. Otherwise the entropy is given by,

$$\lim_{\substack{N\to\infty \\ \text{otherwise}}} S_N^{\text{vN}} = S_1\left( \frac{N}{4(N-2)}, \frac{N}{4(N-2)}, \frac{N-4}{4(N-2)}, \frac{N-4}{4(N-2)} \right)$$
$$= \frac{5}{2}\ln 2. \tag{79}$$

We plot $S_l^{\text{vN}}$ vs $l/L$ in Fig.6 for different momenta. We see, for special momenta (such as $k = 0$), $S_l^{\text{vN}}$ approaches the value $2\ln 2$ as $l \to N$, whereas for all other momenta, it approaches $2.5\ln 2$. These states are thus found to be of area law.

We now consider the $AB$ solutions [Eq. (25)]. Following the same recipe as before, we write the final expressions of eigenvalues of half-chain RDM for even $p$ (where $k = p\pi/N$). In this special case, $\rho_{N,0} = 1/4$. $\rho_{N,1}$ has the following six non-zero eigenvalues: $1/[8(2+\cos(2k))]$ of degeneracy 4, and $[1+\cos(2k)]/[4(2+\cos(2k))]$ of degeneracy 2. The only non-zero eigenvalue of $\rho_{N,2}$ must be the same as $\rho_{N,0}(= 1/4)$. Unlike the previous case, the eigenvalues of RDM and hence the $S_N^{\text{vN}}$ is fixed for a fixed $k$ (does not change with $N$) and shows a maximum ($\approx 2.8\ln 2$) at $k = \pi/3$ [see Fig. 7(a)]. The values of $S_N^{\text{vN}}$ for odd $p$ (for a given $k$) asymptotically touches the corresponding values for even $p$ from below [see Fig. 7(b)]. Thus, these states, though have slightly more entanglement than the previous ones, still exhibit area law.

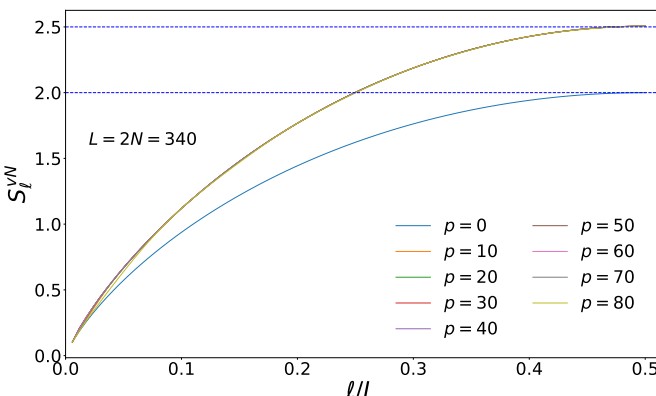

FIG. 6. $S_l^{\mathrm{vN}}$ (in units of $\ln 2$) vs $l/L$ for the "cosine" family of $AA - BB$ type two-magnon states at different momenta $k = p\pi/N$, $N = 170$. Dashed lines represent the asymptotic values of $S_N^{\mathrm{vN}}$ ($2\ln 2$ and $(5/2)\ln 2$).

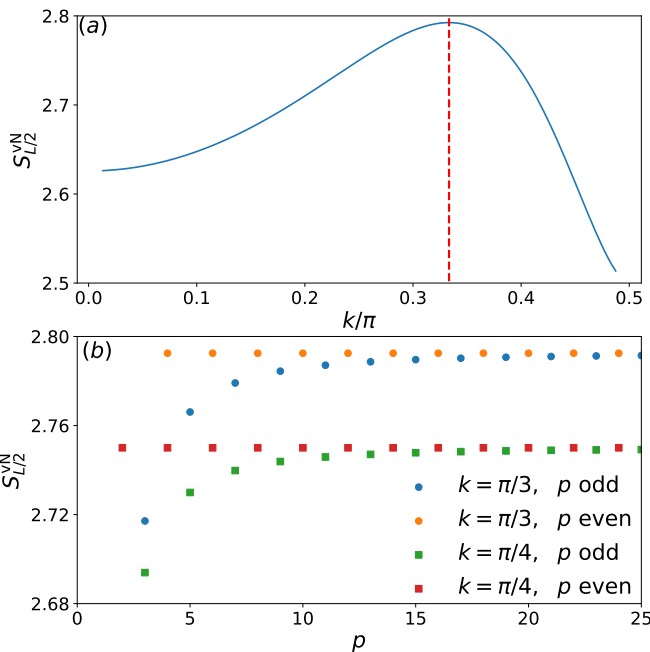

FIG. 7. (a) $S_N^{\mathrm{vN}}$ (in units of $\ln 2$) vs $k/\pi$ for the $AB$ type two-magnon solutions. The red dashed line indicates the position of the maxima. (b) $S_N^{\mathrm{vN}}$ (in units of $\ln 2$) vs $p$ (where $k = p\pi/N$) for $k = \pi/4, \pi/3$.

### B. Three-magnon states

The three-magnon states also exhibit area-law scaling albeit with a higher ($O(1)$) value of entanglement than the two-magnon states. The full analytical calculation of entanglement for any bipartition is difficult and we only attempt to calculate $S_N^{\mathrm{vN}}$ numerically. The calculation can be further simplified by observing that the eigenvalues of $\rho_{N,3}$ and $\rho_{N,2}$ are the same as the eigenvalues of $\rho_{N,0}$ and $\rho_{N,1}$ respectively. We plot $S_N^{\mathrm{vN}}$ vs $k/\pi$ for four different family of three-magnon states in Fig. 8(a). The

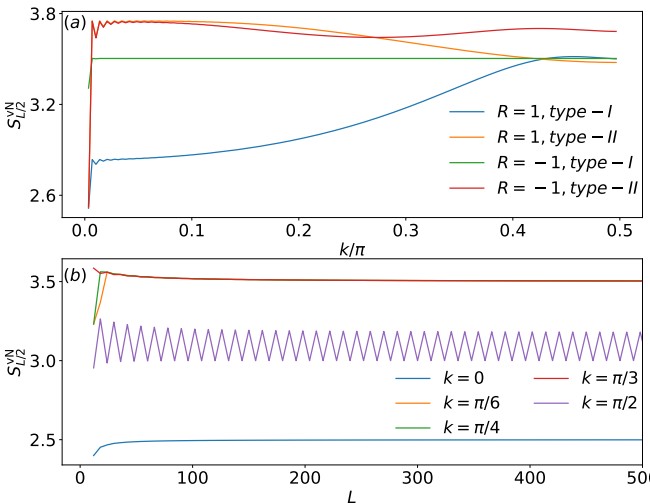

FIG. 8. (a) $S_N^{\mathrm{vN}}$ vs $k/\pi$ for different families of three-magnon states where type-I(II) states are descendants (not descendants) of corresponding two-magnon states. $N = 288$. (b) $S_N^{\mathrm{vN}}$ vs $N$ for the $R = -1$, type-I states at different momenta.

$S_N^{\mathrm{vN}}$ show dispersion with $k$ for all the families except the $R = -1$ three-magnon states which are direct descendants of $AA - BB$ type two-magnon states (which were also found to be dispersion-less in the previous section). The value of $S_N^{\mathrm{vN}}$ (3.5) for this particular class of three-magnon states for $k \in [0, \pi/2]$ is exactly $\ln 2$ higher than the corresponding two-magnon states. This is due to the fact that the spin-lowering operator, connecting the two classes of states, can be represented as a matrix product operator (MPO) with bond-dimension 2 as follows,

$$S_{\mathrm{tot}}^- = \sum_{i=1}^{L} S_i^- = W^{[1]} W^{[2]} W^{[3]} \cdots W^{[L]}, \qquad (80)$$

where,

$$W^{[1]} = \begin{pmatrix} \mathbb{I} & S_1^- \end{pmatrix} \; ; \; W^{[i]} = \begin{pmatrix} \mathbb{I} & S_i^- \\ \mathbf{0} & \mathbb{I} \end{pmatrix} \; ; \; W^{[L]} = \begin{pmatrix} S_L^- \\ \mathbb{I} \end{pmatrix}. \tag{81}$$

The action of this MPO on the matrix product state MPS corresponding to the $AA - BB$ type two-magnon states doubles the bond dimension of the corresponding MPS for any $k$, injecting one unit of entanglement in the thermodynamic limit. The value of $S_N^{\mathrm{vN}}$ for $k = 0$ (2.5) and $\pi/2$ (found to oscillate between two values for odd and even $N$) is different than the rest of the momenta. The value of $S_N^{\mathrm{vN}}$ quickly approaches (with $N$) their thermodynamic value which is a signature of a very small correlation length of these states (see Fig. 8).

## VII.   CONCLUSION

In summary, we have studied aspects of the spectrum of the non-integrable alternating Heisenberg chain, revealing its partially integrable structure. The model is a close cousin of the spin-1 Haldane chain and is the spin analog of the Su-Schrieffer-Heeger chain. Focusing on zero energy eigenstates, we find that the GBA explains all two-magnon states for an arbitrary number of unit cells $N$, both even and odd, and a large number of three-magnon zero energy states (all for even $N$ and a large fraction for odd $N$). We have provided closed-form analytic expressions for these states and for their entanglement entropy. It is interesting to note how multi-magnon wave functions can achieve the zero energy condition by either avoiding each other completely by never being nearest neighbors, thereby avoiding "collisions" (an effective way of satisfying the constraint equations) or alternatively by destructive interference. We note that such states can also be prepared with very small resources on near-term quantum computers [78].

For a higher number of magnons, we have shown evidence for a GBA representation for some zero-energy eigenstates. This was carried out with the numerical-GBA, a procedure to systematically introduce plane wave basis wave functions, in order to diagonalize the square of the Hamiltonian in a reduced basis, after appropriately orthonormalizing the basis functions and removing any linear dependences. The procedure is perfectly general and could be used for other Hamiltonians where a partially integrable structure is possibly present.

More generally, our work shows that a large number of atypical eigenstates, at the heart of quantum many-body scars, exist in the alternating Heisenberg chain. The Bethe states can be considered as "few body" scars and it is conceivable that this partially integrable structure persists for a large number of magnons. Exact volume-law-entangled states have been recently found in the alternating Heisenberg chain [77], suggesting this possibil-ity. It would also be interesting to see how our picture of paired magnons and exact states translates to higher dimensional systems where the Bethe ansatz does not directly apply. For example, for the alternating Heisenberg kagome (with three sublattices), in addition to two dispersive bands (with $\pm E(k)$, which are the equivalent of $\pm \sin k$ for the alternating chain), there is also an exact zero energy flat band [79]. We intend to explore these and related questions elsewhere.

## ACKNOWLEDGMENTS

We thank P. Schlottmann for discussions on the Bethe ansatz technique and for his encouragement. We also thank P. Sharma for pointing us to relevant references. R. M. and H. J. C. acknowledge funding from National Science Foundation Grant No. DMR 2046570 and the National High Magnetic Field Laboratory. The National High Magnetic Field Laboratory is supported by the National Science Foundation through NSF/DMR-1644779 and DMR-2128556 and the state of Florida. A. P., B. M., and M. S. were funded by the European Research Council (ERC) under the European Union's Horizon 2020 research and innovation programme (Grant Agreement No. 853368). B. M. was also funded by DST, Government of India via the INSPIRE Faculty programme. C. J. T. is supported by an EPSRC fellowship (Grant Ref. EP/W005743/1).

## Appendix A: Proof of linear dependence of solutions

When considering the $AA - BB$ family of zero energy states for two magnons, for even $N$, we stated that the wave function for $p = \frac{N}{2}$ ($k = \pi/2$) is linearly dependent on the other eigenmodes. Here, with the help of a mathematical identity, we provide a rigorous justification for this statement.

We use the general result,

$$\sum_{p=0}^{M-1} \cos(a + pd) = \begin{cases} M \cos a & \text{if } \sin\left(\frac{d}{2}\right) = 0, \\ \dfrac{\sin(\frac{Md}{2})}{\sin \frac{d}{2}} \cos\left(a + \frac{(M-1)d}{2}\right) & \text{otherwise.} \end{cases} \tag{A1}$$

Set $M = N/2$ and $a = 0$ and $d = \frac{\pi(2n-2m)}{N}$, where $n, m$ are arbitrary integers restricted to $0 \leq n, m < N$. Since $\sin(\frac{\pi(2n-2m)}{2N}) \neq 0$ for any integer $N$, we obtain

$$\sum_{p=0}^{(N/2)-1} \cos\left(\frac{p\pi(2n-2m)}{N}\right) = \frac{\sin\left(\frac{\pi(n-m)}{2}\right)}{\sin\left(\frac{\pi(n-m)}{N}\right)} \cos\left(\left(\frac{N}{2} - 1\right)\frac{\pi(n-m)}{N}\right) \tag{A2}$$

$$= \frac{\sin\left(\frac{\pi(n-m)}{2}\right)}{\sin\left(\frac{\pi(n-m)}{N}\right)} \left[\cos\left(\frac{\pi(n-m)}{2}\right)\cos\left(\frac{\pi(n-m)}{N}\right) + \sin\left(\frac{\pi(n-m)}{2}\right)\sin\left(\frac{\pi(n-m)}{N}\right)\right].$$

We simplify this expression further. Consider the case of even $(n-m)$. Then $\sin(\pi(n-m)/2) = 0$ and thus the entire series sums to zero,

$$\sum_{p=0}^{(N/2)-1} \cos\left(\frac{p\pi(2n-2m)}{N}\right) = 0 \quad \text{for even } (n-m). \tag{A3}$$

For odd $(n-m)$, $\cos(\pi(n-m)/2) = 0$, and thus

$$\sum_{p=0}^{(N/2)-1} \cos\left(\frac{p\pi(2n-2m)}{N}\right) = \sin^2\left(\frac{\pi(n-m)}{2}\right) = 1 \quad \text{for odd } (n-m). \tag{A4}$$

Summarizing these two results in a single equation,

$$\sum_{p=0}^{(N/2)-1} \cos\left(\frac{p\pi(2n-2m)}{N}\right) = \frac{1 - \cos(\pi(n-m))}{2}. \tag{A5}$$

We recast this slightly differently by noting that $\cos(\pi(n-m)) = \cos(\frac{N}{2}\frac{\pi(2n-2m)}{N})$ and $1 = \cos(0\frac{\pi(2n-2m)}{N})$,

$$\cos\left(\frac{N}{2}\frac{\pi(2n-2m)}{N}\right) = \cos\left(0\frac{\pi(2n-2m)}{N}\right) - 2\sum_{p=0}^{(N/2)-1} \cos\left(\frac{p\pi(2n-2m)}{N}\right). \tag{A6}$$

## Appendix B: Checking other momentum combinations in the generalized Bethe ansatz

The solution of the hopping equations leads to a relationship between the momenta that appear in the GBA. In this appendix, we explain why the case of $k_1 = k_2 \neq 0$ does not yield zero energy Bethe solutions. We do not incorporate any constant terms because they are associated with total momentum zero, i.e. a different symmetry sector.

For $k_1 = k_2 = k$, the ansatz for the two-magnon wave function is,

$$a_{n,m} = \alpha_{f(n),f(m)} e^{ik(n+m)}, \tag{B1}$$

where $f(n) = A$ for $n$ on an even sublattice and $f(n) = B$ for $n$ on an odd sublattice. We have from the hopping equations,

$$\alpha_{AA} = +\alpha_{BB}, \qquad \alpha_{AB} = -\alpha_{BA}. \tag{B2}$$

Plugging the resultant wave function into the two constraint equations we get,

$$-2\alpha_{AB} e^{ik(4n-1)} = \alpha_{AA} e^{ik(4n-2)} + \alpha_{AA} e^{ik(4n)}, \tag{B3a}$$
$$2\alpha_{AB} e^{ik(4n+1)} = \alpha_{AA} e^{ik(4n)} + \alpha_{AA} e^{ik(4n+2)}. \tag{B3b}$$

which implies that

$$-\alpha_{AB} = \alpha_{AA} \cos k \quad \text{and} \quad +\alpha_{AB} = \alpha_{AA} \cos k. \tag{B4}$$

These conditions are inconsistent with one another unless $k = \frac{\pi}{2}$ and $\alpha_{AB} = 0$. (This latter case yields an $AA - BB$ solution which was considered elsewhere.) Thus, the case of $k_1 = k_2$ admits no new solutions for any $N$.

## Appendix C: The search for three-magnon exact zero energy states using the generalized Bethe ansatz state with momenta $k_1 = k_2 = k$, $k_3 = 0$ and $k_1 = k_2 = k + \frac{\pi}{2}$ and $k_3 = 0$

We use the superscripts (1) and (2) to denote momenta $k^{(1)} \equiv k = \frac{2\pi}{2N}\mathbb{Z}$ and $k^{(2)} = k + \frac{\pi}{2}$ which satisfy $e^{2ik^{(1)}N} = 1$ and $e^{2ik^{(2)}N} = -1$ respectively. The GBA of the proposed state has three-magnon wave function coefficients,

$$a_{n,m,l} = \sum_{j=1}^{2} a_{n,m,l}^{(j)} = \sum_{j=1}^{2} \left(\alpha_{f(n),f(m),f(l)}^{(j)} e^{ik^{(j)}(n+m)} + \beta_{f(n),f(m),f(l)}^{(j)} e^{ik^{(j)}(n+l)} + \gamma_{f(n),f(m),f(l)}^{(j)} e^{ik^{(j)}(m+l)}\right), \tag{C1}$$

where $\alpha$, $\beta$ and $\gamma$ are Ansatz parameters, independent for each momentum and sublattice index. The number of free parameters is reduced by solving the hopping equations and applying periodic boundary conditions to obtain,

$$a^{(j)}_{2n,2m,2l} = A^{(j)}e^{ik^{(j)}(2n+2m)} + A^{(j)}e^{2ik^{(j)}N}e^{ik^{(j)}(2n+2l)} + A^{(j)}e^{ik^{(j)}(2m+2l)}, \tag{C2a}$$

$$a^{(j)}_{2n,2m,2l+1} = B^{(j)}e^{ik^{(j)}(2n+2m)} - C^{(j)}e^{2ik^{(j)}N}e^{ik^{(j)}(2n+2l+1)} + C^{(j)}e^{ik^{(j)}(2m+2l+1)}, \tag{C2b}$$

$$a^{(j)}_{2n,2m+1,2l} = C^{(j)}e^{ik^{(j)}(2n+2m+1)} + B^{(j)}e^{2ik^{(j)}N}e^{ik^{(j)}(2n+2l)} - C^{(j)}e^{ik^{(j)}(2m+2l+1)}, \tag{C2c}$$

$$a^{(j)}_{2n,2m+1,2l+1} = D^{(j)}e^{ik^{(j)}(2n+2m+1)} - D^{(j)}e^{2ik^{(j)}N}e^{ik^{(j)}(2n+2l+1)} + A^{(j)}e^{ik^{(j)}(2m+2l+2)}, \tag{C2d}$$

$$a^{(j)}_{2n+1,2m,2l} = -C^{(j)}e^{ik^{(j)}(2n+2m+1)} + C^{(j)}e^{2ik^{(j)}N}e^{ik^{(j)}(2n+2l+1)} + B^{(j)}e^{ik^{(j)}(2m+2l)}, \tag{C2e}$$

$$a^{(j)}_{2n+1,2m,2l+1} = -D^{(j)}e^{ik^{(j)}(2n+2m+1)} + A^{(j)}e^{2ik^{(j)}N}e^{ik^{(j)}(2n+2l+2)} + D^{(j)}e^{ik^{(j)}(2m+2l+1)}, \tag{C2f}$$

$$a^{(j)}_{2n+1,2m+1,2l} = A^{(j)}e^{ik^{(j)}(2n+2m+2)} + D^{(j)}e^{2ik^{(j)}N}e^{ik^{(j)}(2n+2l+1)} - D^{(j)}e^{ik^{(j)}(2m+2l+1)}, \tag{C2g}$$

$$a^{(j)}_{2n+1,2m+1,2l+1} = B^{(j)}e^{ik^{(j)}(2n+2m+2)} + B^{(j)}e^{2ik^{(j)}N}e^{ik^{(j)}(2n+2l+2)} + B^{(j)}e^{ik^{(j)}(2m+2l+2)}, \tag{C2h}$$

where we have now defined eight unknown parameters $A^{(j)}, B^{(j)}, C^{(j)}, D^{(j)}$ for $j = 1, 2$. We will determine the values of these parameters that satisfy the zero energy constraint equations.

Consider first the two constraints arising from configurations of three magnons on consecutive sites. We get two equations,

$$C^{(2)} = +i(B^{(2)} - B^{(1)})\cos k, \tag{C3a}$$

$$D^{(2)} = -i(A^{(2)} + A^{(1)})\cos k. \tag{C3b}$$

Next, consider the constraint equation corresponding to two magnons that are nearest neighbors and the third magnon (not a nearest neighbor of either) on an even sublattice. These yield three equations,

$$D^{(2)} - iD^{(1)} = -B^{(2)}\sin k - iB^{(1)}\cos k, \tag{C4a}$$

$$D^{(1)} = (A^{(1)} - B^{(1)})e^{ik}, \tag{C4b}$$

$$A^{(2)} = i(D^{(2)} - C^{(2)})e^{-ik}. \tag{C4c}$$

When the third magnon is instead on an odd sublattice we get,

$$C^{(2)} - iC^{(1)} = -A^{(2)}\sin k - iA^{(1)}\cos k, \tag{C5a}$$

$$C^{(1)} = (B^{(1)} - A^{(1)})e^{-ik}, \tag{C5b}$$

$$B^{(2)} = i(D^{(2)} - C^{(2)})e^{ik}. \tag{C5c}$$

This system of eight equations (C3a)–(C5c) has a unique solution (up to a multiplicative factor) which we find to be,

$$A^{(2)} = B^{(2)} = C^{(1)} = D^{(1)} = 0, \tag{C6a}$$

$$A^{(1)} = B^{(1)} = 1, \tag{C6b}$$

$$C^{(2)} = D^{(2)} = (-i\cos k). \tag{C6c}$$

These parameters yield a three-magnon wave function that is the spin-lowered descendant of the two-magnon state found in Sec. III D.

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

579 (2017).

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

Lett. **127**, 060602 (2021).