# Peer review of "Exact generalized Bethe eigenstates of the non-integrable alternating Heisenberg chain"

_SciPost Physics_

## Round 1 · Referee Report · Anonymous (Referee 1) · 2025-4-10

Report

In the present manuscript, the authors explore the spin 1/2 alternating ferromagnetic-antiferromagnetic Heisenberg chain and identify exactly solvable states. They introduce a generalized Bethe Ansatz (GBA) calculation capable of generating many-particle zero-energy scar states, which they systematically analyze both theoretically and numerically. While the manuscript presents comprehensive discussions, several key points require further clarification for publication:

The manuscript rigorously presents wavefunctions and GBA equations for one-, two-, and three-magnon sectors (sections II-IV), and solves the GBA numerically up to the four-magnon sector (section V). However, it remains unclear whether these scar states persist in arbitrarily large N-magnon sectors—an assertion that warrants explicit proof or argumentation.

The authors assert area-law entanglement properties for the studied scar states observed in few-magnon states. Given the thermodynamic limit, where all zero particle density states exhibit area-law entanglement trivially, it is crucial to investigate how the entanglement scales in scar states with finite magnon densities. Specifically, understanding the scaling behavior towards the thermodynamic limit, including potential deviations from area law such as critical or volume-law entanglement, would enhance the manuscript's completeness.

As scar states are distinguished by their violation of the Eigenstate Thermalization Hypothesis (ETH), it would be insightful to explore observable quantities—like magnetization—that differentiate scar states from thermal equilibrium states. This analysis would further elucidate the physical implications and experimental relevance of the identified scar states.

Overall, with these modifications and clarifications, the manuscript presents a valuable contribution to the field and deserves publication.

Recommendation

Ask for minor revision

---

## Round 1 · Referee Report · Anonymous (Referee 2) · 2025-4-17

Strengths

see report

Weaknesses

see report

Report

Referee Report on the Manuscript:

"Exact generalized Bethe eigenstates of the non-integrable alternating Heisenberg chain"

I have read the manuscript and have several substantive concerns regarding its
claims and conclusions.

The authors assert the existence of "partial integrability" in the alternating
Heisenberg chain (AHC) for a larger number of magnons. Upon closer
examination, what is actually demonstrated is that in spin models with
conserved particle number (i.e., magnon number), one- and two-particle states
can be constructed in accordance with standard quantum mechanical
principles. The two-magnon states presented are simply conventional
two-particle scattering states. This behavior is generic and does not
constitute a signature of integrability.

In the case of three magnons, the authors construct specific states as
superpositions of plane waves, with additional matching conditions. Similar
constructions are attempted for higher numbers of magnons, but all are
restricted to the special case of zero-energy states. Although these states
formally resemble Bethe ansatz-type wavefunctions, the results apply only to
narrow and atypical subsets of the full spectrum.

The authors do succeed in analytically constructing a large number of
zero-energy states, consistent with prior numerical diagonalizations. However,
these zero-energy states lie in the "middle" of the many-body spectrum. The
manuscript does not convincingly articulate why these states are of particular
physical interest or relevance.

Furthermore, the manuscript contains a significant misconception. The authors
state:

“The key reason for the success of the Bethe ansatz is the tractability of the
momenta of the particles in an integrable model – when two particles collide,
their momenta after the collision are simply exchanged.”

This characterization is misleading. The simple exchange of momenta upon
two-particle scattering is a common feature of many quantum systems, not
exclusive to integrable ones. The true hallmark of integrability is the
absence of intrinsic many-particle scattering — that is, the factorization of
the full scattering matrix into two-particle S-matrices. Importantly, the
Yang-Baxter equation provides a necessary (but not sufficient) condition for
such factorization.

Given these points, I do not consider the manuscript suitable for publication
in SciPost in its current form.

Recommendation

Reject

---

## Round 1 · Referee Report · Anonymous (Referee 3) · 2025-5-7

Strengths

-makes new direction into quantitative description of zero-states of non-integrable systems (a potential new avenue for research)

-it provides a substantial amount of useful data

Weaknesses

-not clear what is really the relationship to coordinate Bethe Ansatz

-the explorations seem quite ad hoc, lack more systematic approaches

Report

I think this is an interesting and intriguing manuscript.

It attempts to explain the curious set of so-called zero-energy eigenstates in the non-integrable alternating spin-1/2 Heisenberg chain. The key idea is to use an ansatz inspired by coordinate Bethe ansatz, i.e. a linear superposition of multi-magnon states. Then the sectors with small number of magnons are quite systematically studied, the states are counted sector by sector and their structure is explained case by case. Clearly, the structure of the propblem is largely affected by the non-abelian (SU(2)) symmetry of the model, i.e. zero-energy eigenstates can be organized into SU(2) multiplets.

In terms of organizing the case-by-case calculations and presenting the corroborating empirical data, the manyscript is clear. However, I have a problem with the "big picture".

My main problem with the manuscript is that I can't identify a clear link with the Bethe Ansatz of integrable systems. I would prefer to call the authors' ansatz simply a multi-magnon plane wave superpositon ansatz. As far as I understand, the key to integrability is the factorization of multi-particle (or multi-magnon) scattering into a product of 2-particle scatterings in irrelevant order. This is the celebrated Yang-Baxter equation and results in simple and compact structure of Bethe ansatz. This seems to be missing here, so the term generalized BA seems very missleading. What do the authors mean with "partial integrability"? I don't think there is any integrability here, but perhaps there is a different interesting algeraic structure (yet to be discovered) which would explain the proliferation of zero-energy eigenstates and their "plane-wave" structure (e.g magnon pairings. etc). This are all interesting observations worth publishing, but I think the manuscript is overselling in interpretation.

I would thus suggest the authors to rewrite the manuscript in this respect, or provide clear evidence for their claim of partial integrability in terms of some kind of partial Yang-Baxter equation (partial factorizability of 2-body scatterings?).

Requested changes

  • In introduction, the authors claim that XXZ systems have been shown to exhibit weak ergodicity breaking. But XXZ is an intergable system, so perhaps they should be more precise there?

  • I think the term "super-spin" deserves a definition.

  • The authors speculate that they GBA procedure can be used beyond zero-energy sectors. In light of what I elaborated above I have doubts about this claim. I would thus ask the authors to either provide some example or better motivation for this claim, or remove it.

Recommendation

Ask for major revision

---

## Editorial Decision

awaiting_resubmission